# Negative frequency-dependent interactions can underlie phenotypic heterogeneity in a clonal microbial population

David Healey[1], Kevin Axelrod[2] & Jeff Gore[3,*]

## Abstract

Genetically identical cells in microbial populations often exhibit a remarkable degree of phenotypic heterogeneity even in homogenous environments. Such heterogeneity is commonly thought to represent a bet-hedging strategy against environmental uncertainty. However, evolutionary game theory predicts that phenotypic heterogeneity may also be a response to negative frequency-dependent interactions that favor rare phenotypes over common ones. Here we provide experimental evidence for this alternative explanation in the context of the well-studied yeast GAL network. In an environment containing the two sugars glucose and galactose, the yeast GAL network displays stochastic bimodal activation. We show that in this mixed sugar environment, GAL-ON and GAL-OFF phenotypes can each invade the opposite phenotype when rare and that there exists a resulting stable mix of phenotypes. Consistent with theoretical predictions, the resulting stable mix of phenotypes is not necessarily optimal for population growth. We find that the wild-type mixed strategist GAL network can invade populations of both pure strategists while remaining uninvasible by either. Lastly, using laboratory evolution we show that this mixed resource environment can directly drive the *de novo* evolution of clonal phenotypic heterogeneity from a pure strategist population. Taken together, our results provide experimental evidence that negative frequency-dependent interactions can underlie the phenotypic heterogeneity found in clonal microbial populations.

**Keywords** ecology; evolution; frequency dependence; phenotypic heterogeneity; stochastic gene expression
**Subject Categories** Evolution; Microbiology, Virology & Host Pathogen Interaction; Quantitative Biology & Dynamical Systems
**Mol Syst Biol. (2016) 12: 877**

See also: **DA Charlebois & G Balázsi** (August 2016)

## Introduction

Stochastic gene expression is ubiquitous in microbial populations (Thattai & van Oudenaarden, 2001; Elowitz *et al*, 2002; Avery, 2006; Maamar *et al*, 2007). In some cases, stochastic gene expression results in the coexistence of distinct phenotypes among genetically identical cells (Becskei *et al*, 2001; Dubnau & Losick, 2006). Examples include persistence (Balaban *et al*, 2004), competence (Suel *et al*, 2006; Maamar *et al*, 2007), and sporulation (Veening *et al*, 2005) in bacteria. Such phenotypic heterogeneity is an evolvable trait whose quantitative parameters can be tuned by the architecture and properties of the underlying gene network (Raser & O'Shea, 2004; Acar *et al*, 2008; Beaumont *et al*, 2009; Cagatay *et al*, 2009). This raises the question of what adaptive advantage might be conferred to cells that implement stochastic decision-making (Wolf *et al*, 2005a; Blake *et al*, 2006; Bishop *et al*, 2007; Fraser & Kaern, 2009; Sharma *et al*, 2010; Pisco *et al*, 2013; Viney & Reece, 2013). There are theoretically multiple survival strategies that could be implemented by phenotypic heterogeneity (Perkins & Swain, 2009; Wylie *et al*, 2010; de Jong *et al*, 2011; Adami *et al*, 2012). Most commonly, heterogeneity in clonal microbial populations is thought to be a bet-hedging response to environmental uncertainty; populations that stochastically adopt a range of phenotypes can gain a fitness advantage if the environment shifts unexpectedly (Kussell *et al*, 2005; Acar *et al*, 2008; Veening *et al*, 2008; Beaumont *et al*, 2009; Simons, 2011; Levy *et al*, 2012; Viney & Reece, 2013; Weinberger & Weinberger, 2013; Pradhan & Chatterjee, 2014). For example, bacteria may, at some frequency, stochastically adopt a dormant or slow-growing "persister" state, which has reduced fitness in abundant resources, but which is more likely to survive in the event of unexpected catastrophic environmental stress (Kussell *et al*, 2005; Lennon & Jones, 2011).

Another explanation, which has been generally overlooked in describing clonal microbial populations, is that phenotypic heterogeneity may be driven by negative frequency-dependent interactions between the relevant phenotypes. Negative frequency-dependent selection occurs when the relative fitness of a phenotype decreases

1   Department of Biology, Massachusetts Institute of Technology, Cambridge, MA, USA
2   Graduate Program in Biophysics, Harvard University, Cambridge, MA, USA
3   Department of Physics, Massachusetts Institute of Technology, Cambridge, MA, USA
    *Corresponding author. Tel: +1 6172534829; E-mail: gore@mit.edu

with its increasing prevalence. Two phenotypes are said to be *mutually invasible* when frequency dependence is such that each phenotype can invade the other when rare. Negative frequency dependence is known to stabilize genetic diversity in microbial populations (Xavier & Foster, 2007; Gore *et al*, 2009; Sanchez & Gore, 2013; Cordero & Polz, 2014); however, a genetically identical population can theoretically achieve the same stable mix of phenotypes if each individual randomizes between the phenotypes with appropriate probabilities. The field of evolutionary game theory originally modeled these probabilistic strategies in the context of the hawk–dove model of animal conflict (Maynard Smith, 1982), where neither deterministic pure strategy (i.e. "always hawk" or "always dove") is evolutionarily stable, and the only evolutionarily stable strategy is a probabilistic choice between pure strategies, called a mixed strategy. Although theory predicts that evolutionarily stable mixed strategies can result from negative frequency-dependent interactions (Wolf *et al*, 2005b; de Jong *et al*, 2011; Adami *et al*, 2012), as yet there is a lack of experimental evidence for such interactions as a driving force of phenotypic heterogeneity in clonal microbial populations.

It is difficult, if not impossible, to know the relevant evolutionary history of natural microbial populations, and by extension the reasons why they may have evolved phenotypic heterogeneity. This is further complicated by the fact that multiple evolutionary drivers may apply to a given situation. However, both uncertain environments and frequency-dependent interactions are alone sufficient for a population to evolve phenotypic heterogeneity, and each makes unique and experimentally observable predictions regarding the fitness dynamics between the represented phenotypes. For stable mixed strategies, the primary defining characteristic is that the corresponding pure strategies are mutually invasible: each can invade the other when rare, driving the overall population toward a stable mix of phenotypes in which each phenotype has equal fitness. Pure strategies are not mutually invasible in a bet-hedging scenario. Additionally, while bet hedging increases the mean growth of the population relative to phenotypically homogenous strategies, the stable mix resulting from negative frequency-dependent selection merely equalizes the fitness of the phenotypes and is not necessarily optimal for population growth (Parker & Smith, 1990; Quenette & Gerard, 1992; see Appendix Text S1 for a more thorough introduction to bet-hedging and frequency-dependent evolutionary games).

## Results

To study evolutionarily stable mixed strategies in the laboratory, we investigated the decision of the budding yeast *S. cerevisiae* regarding which carbon source to consume. Yeast prefers the sugar glucose, but when glucose is limited, yeast can consume other carbon sources (Gancedo, 1998). The well-studied yeast GAL network contains the suite of genes needed to metabolize the sugar galactose. However, rather than follow the canonical diauxic growth pattern of consuming glucose before activating the GAL network, yeast can still activate the GAL genes in the presence of glucose, depending on the ratio of galactose to glucose in the media (Appendix Fig S1; Escalante-Chong *et al*, 2015). Furthermore, the response is not always uniform across the population. Using a strain

containing YFP driven from a *GAL1* promoter, we confirmed via flow cytometry that there exists a wide range of mixed glucose/ galactose environments wherein some clonal W303 yeast cells activate the GAL network, while others do not (Appendix Fig S1; Acar *et al*, 2005; Venturelli *et al*, 2015; Wang *et al*, 2015). Consistent with observations from previous studies (Escalante-Chong *et al*, 2015; Venturelli *et al*, 2015), we observe that in the presence of significant levels of glucose and galactose (at least 0.03% of each), this initial bimodal activation is relatively stable over time while both sugars remain abundant (Appendix Fig S4B), although we observe a modest rate of stochastic switching between the states after the initial stochastic bimodal activation (Appendix Fig S5).

From the perspective of evolutionary fitness, activation of the GAL network in the presence of glucose represents a cost-benefit tradeoff: It may offer some benefits to the cell in consuming galactose, but expression of the GAL genes also imposes a significant metabolic cost (Venturelli *et al*, 2015; Wang *et al*, 2015; see also Appendix Fig S2D for details in the present context). Similar tradeoffs in metabolic networks have been characterized previously (Friesen *et al*, 2004; Lambert & Kussell, 2014; New *et al*, 2014; Solopova *et al*, 2014; Gonzalez *et al*, 2015; Lin & Kussell, 2016), but whether the tradeoff is frequency-dependent remains largely unexplored.

Frequency-dependent interactions may be especially relevant to multi-resource environments because of the simple foraging game that exists when resources are limited. For instance, consider an isogenic population that is confronted with a phenotypic decision to specialize in consuming one or the other of two limited food sources, A and B (Fig 1A). We will define two available phenotypes as "specialize in A" or "specialize in B". We will define a *strategy* as some heritable set of instructions about which phenotype to adopt. The more individuals that adopt the strategy "always specialize in A" (a *pure strategy*), the more quickly A will be consumed, reducing payout to individuals who choose that strategy. Hence, if all individuals choose "always specialize in A", a mutant pure strategist that always specializes in B will have an evolutionary advantage and vice versa. Over the course of the resource depletion, the two pure strategies may thus be considered *mutually invasible*, with an equilibrium consisting of a stable mix of the two (Fig 1B). In theory, a clonal population may achieve the same stable mix of phenotypes by adopting a stochastic mixed strategy, wherein each individual adopts one or the other phenotype with some probability, by either making a single random choice or stochastically switching between the two phenotypes. Either way, in that environment, an isogenic population that adopts the stable mixed strategy via phenotypic heterogeneity renders itself uninvasible by a mutant implementing either pure strategy or any other possible stochastic mix between the two.

Of course, this simple foraging game becomes more complicated in the context of a microbial population, which may undergo generations of exponential growth over the course of the resource depletion. In a simple model of a microbial population growing in the presence of two resources, where each phenotype is able to consume one of the two resources (see Appendix Text S1 for model parameters, Appendix Figs S12–S15), we indeed observe negative frequency-dependent selection between the pure strategists and the existence of an evolutionarily stable mix. One result of this simulation is the observation that if individuals of one phenotype must

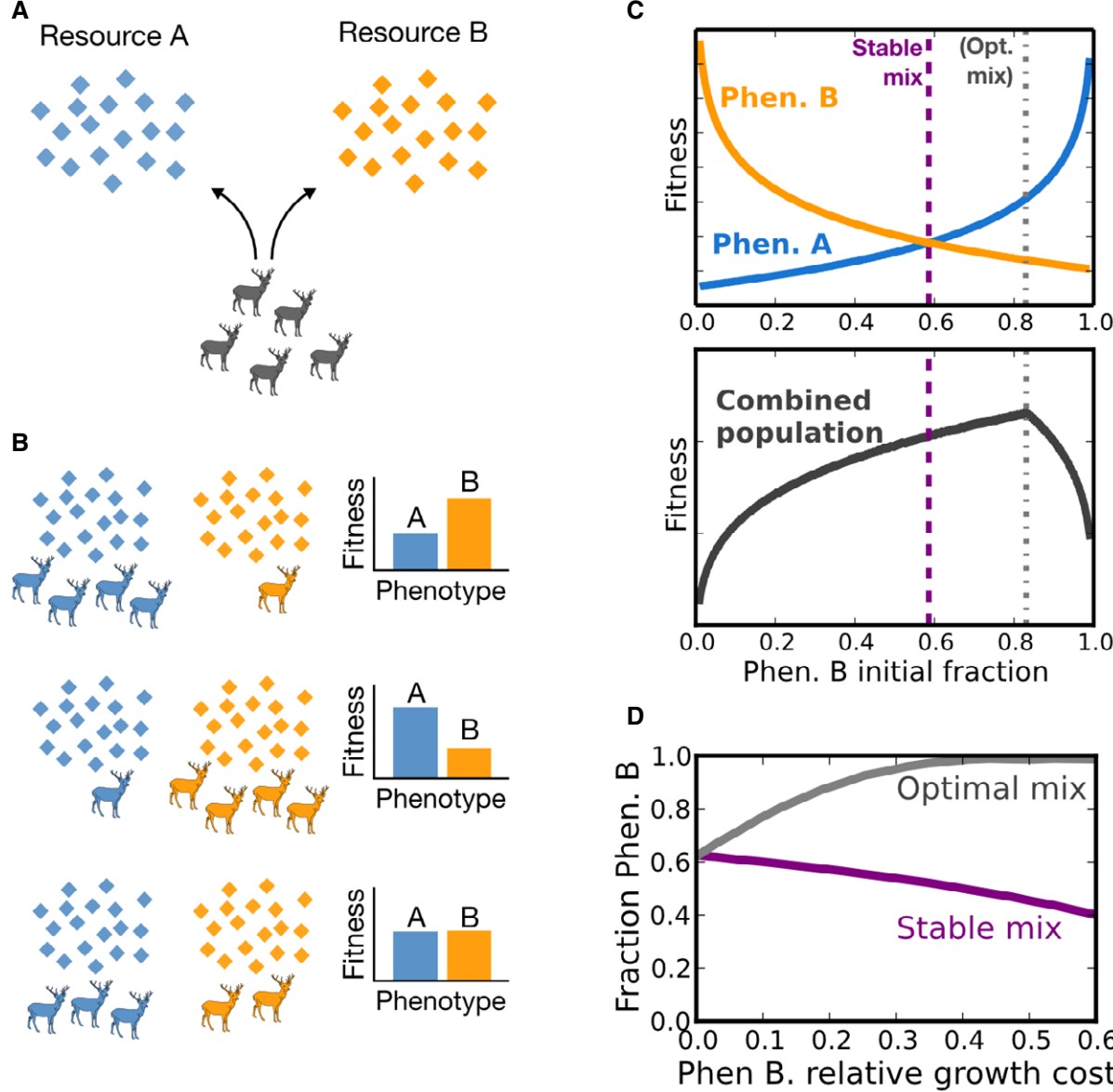

**Figure 1.  Negative frequency-dependent interactions exist in a simple foraging game with multiple resources.**

A   A simple foraging game with a mixed equilibrium: a group of genetically identical foragers encounters two resources: A and (a less preferred resource) B. Each individual must specialize in consuming one or the other. We assume that individuals choose simultaneously and without knowledge of the intentions of other individuals. Because of resource limitations, each individual's fitness is a function of the actions of other individuals.

B   If all other members of the population adopt some pure strategy (e.g., "specialize in resource A"), an individual opting for the opposite pure strategy (e.g., "specialize in resource B") gains a fitness advantage, and vice versa. Hence, a population of pure strategists will always be open to invasion by the opposite pure strategist. A stable equilibrium is reached when the population divides between the two resources such that both phenotypes have equal expected fitness. A genetic strategy that results in this phenotypic heterogeneity is evolutionarily stable. This simple scenario serves to illustrate why we might expect environments with multiple food sources to favor the evolution of mixed strategies.

C   Modeling this simple game in the context of a growing microbial population demonstrates negative frequency-dependent selection between the two phenotypes (A and B) and the existence of the evolutionarily stable mix. Importantly, the stable mix is not identical to the optimal division of labor that would maximize population growth.

D   Indeed, with increasing disparity in growth rates between the phenotypes, the two solution concepts diverge. A higher growth penalty for phenotype B results in a higher phenotype B fraction in the growth-optimal mix, but a lower fraction in the stable one. For details and parameters of the model, see Appendix Text S2 (Appendix Figs S16–S22).

pay a "growth cost" to maintain the phenotype, the stable mix of phenotypes is not identical to the optimal mix that maximizes growth (Fig 1C). Indeed, with increasing growth cost, the two solution concepts diverge (Fig 1D): As the cost incurred by phenotype B increases, the growth-optimal fraction of phenotype B also increases, as the population needs to devote more individuals to this phenotype in order to compensate for the slower growth and resource consumption (population fitness is maximized where both

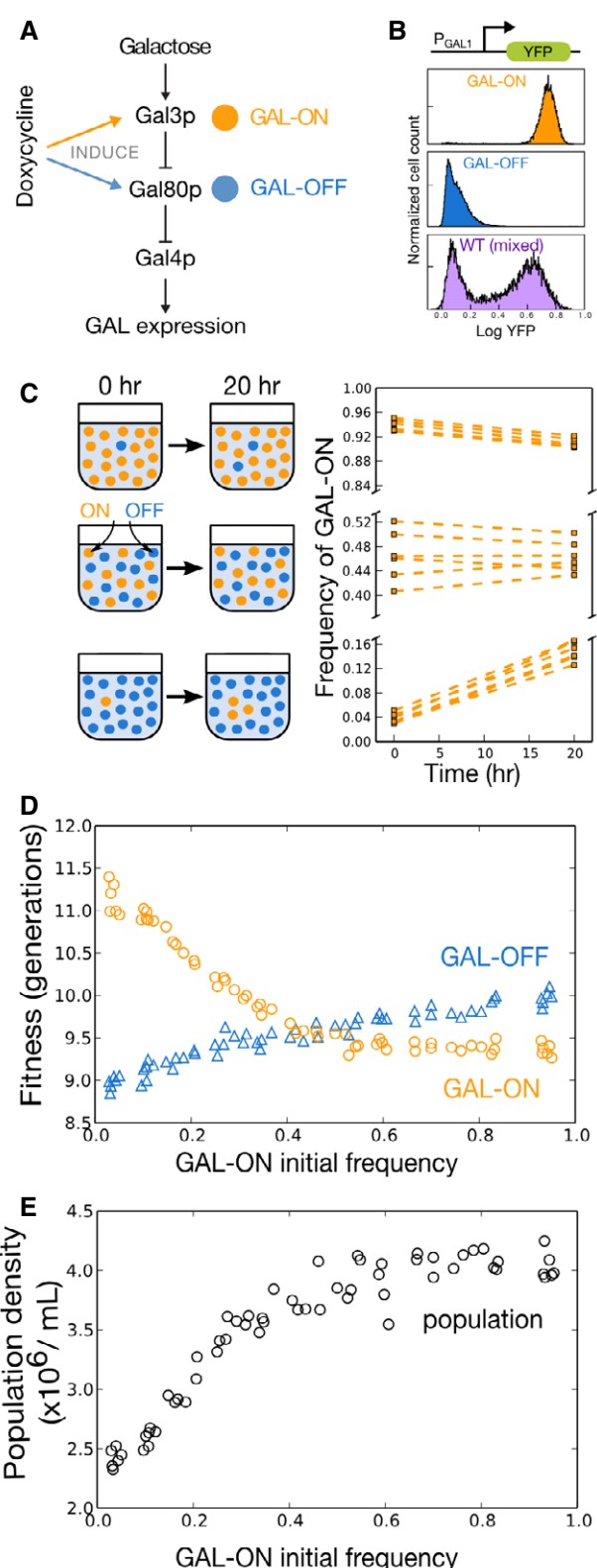

**Figure 2. Yeast GAL network ON and OFF phenotypes are mutually invasible in mixed glucose and galactose.**

A Gene expression in the yeast GAL network is regulated in part by *GAL4*, *GAL80*, and *GAL3* (full network not shown). The wild-type W303 yeast GAL network adopts a mixed strategy in glucose and galactose, but a GAL-OFF pure strategist can be engineered by inducing *GAL80*, whose protein product inhibits GAL expression. Likewise, a GAL-ON pure strategist can be engineered by inducing expression of *GAL3*, which inhibits GAL80 in the presence of galactose.

B YFP driven by a *GAL1* promoter allows for determination of GAL activation states via flow cytometry. GAL activation histograms are shown for the engineered pure strategist strains and for the wild-type W303 yeast GAL network. After incubation for 10 h in a mixed sugar environment (0.03% (w/v) glucose, 0.05% (w/v) galactose, 1 µg/ml doxycycline), GAL-ON and GAL-OFF pure strategists remain unimodally activated and inactivated, respectively, for the GAL network, while the wild-type GAL network exhibits bimodal gene expression. To induce *GAL3* and *GAL80*, cultures were initially incubated in 0.01% glucose and 1 µg/ml doxycycline for 24 h before being transferred to the mixed sugar environment.

C GAL-ON and GAL-OFF phenotypes are mutually invasible in mixed sugar conditions. Population frequency of the *GAL3*-induced (GAL-ON) pure strategists (orange circles) cultured with *GAL80*-induced (GAL-OFF) pure strategists is plotted at the beginning and end of a 20-h incubation in mixed sugars. Six biological replicates of each pure strategist were mixed at high (top panel), intermediate (middle panel), and low (bottom panel) initial GAL-ON fractions. Each pure strategist invades the other when rare. Dashed lines indicate that frequencies are shown at beginning and end to convey the direction of overall frequency change. Actual temporal fitness changes are not linear through the course of the 20-h competition; for more granular temporal fitness dynamics, see Appendix Fig S1.

D Fitness, in number of generations, is shown for the two phenotypes as a function of the population frequency of GAL-ON. Data is shown for sixty cultures and sixty initial fractions representing six biological replicate pairs. The crossing point indicates an evolutionarily stable coexistence of around 40% GAL-ON for this mix of sugars.

E The total population density for the mixed GAL-ON/GAL-OFF populations is plotted against the initial population frequency of GAL-ON. The data are shown for 16 h after incubation, when populations dominated by GAL-ON are already reaching saturating density (~4 × 10⁶/ml), but populations of intermediate or low numbers of GAL-ON cells are still growing. Growth at 16 rather than 20 h is shown because by 20 h the high and intermediate GAL-ON populations have saturated too much to show growth differences between them (see Appendix Fig S7 for final saturating densities). Panels (D) and (E) taken together indicate that the evolutionarily stable mix contains less initial GAL-ON than is growth-optimal from a population standpoint.

Source data are available online for this figure.

Given the bimodal expression of the yeast GAL network in mixed sugar conditions, we sought to probe experimentally whether this phenotypic heterogeneity might be the implementation of an evolutionarily stable mixed strategy in response to frequency-dependent foraging interactions. Since mutual invasibility of phenotypes is the hallmark of such strategies, we began by investigating frequency dependence between GAL-ON and GAL-OFF pure strategies. As a GAL-OFF pure strategist strain, we used a yeast strain whose native *GAL80* (a repressor of the GAL network, Fig 2A) was replaced with a mutant version containing a tet-inducible *TETO2* promoter (Acar *et al*, 2005). As a GAL-ON pure strategist strain, we drove *GAL3* (a repressor of *GAL80*, Fig 2A) from the same promoter. We confirmed that in the range of glucose and galactose concentrations that induce bimodality in wild-type yeast, our doxycycline-induced GAL-OFF and GAL-ON pure strategists are unimodally inactivated and activated, respectively, for GAL gene expression (Fig 2B and Appendix Fig S1).

resources are depleted at the same time). Over the course of the resource depletion, however, at this growth-optimal fraction the individuals with phenotype B have lower fitness than those with phenotype A, making this fraction evolutionarily unstable.

To test for negative frequency dependence between the pure strategists, we mixed six biological replicate pairs of RFP-labeled GAL-OFF and CFP-labeled GAL-ON strains at a total of 60 different initial frequencies, and incubated them in a mixed glucose and galactose environment (0.03% (w/v) glucose and 0.05% galactose, Fig 2C–E) for 20 h, until sugars were consumed. Precise fitness values (in number of divisions or generations undergone by each strategist) were determined by measuring population composition before and after incubation via flow cytometry (Appendix Fig S5). We found that small populations of each pure strategist were indeed able to invade majority populations of the other (Fig 2C). Our experimental yeast populations therefore display mutual invasibility between the two pure strategists. Consistent with mutual invasibility, there was a unique stable equilibrium frequency of GAL-ON cells wherein both pure strategists had equal fitness (Fig 2D). Importantly, we find that the frequency of GAL-ON cells that is evolutionary stable is not the frequency that maximizes population growth, as populations with much higher fractions of GAL-ON cells than the equilibrium population grow to saturating density more quickly than the evolutionarily stable population (Fig 2E).

A more in-depth investigation of the dynamics between the pure strategists indicates that while our system differs in several respects from the simulated models, the negative frequency dependence we observe is indeed related to the depletion of resources in the media. Unlike in the simulated models where each specialist strategy only consumes its niche resource, we find that in the presence of glucose and galactose, both pure strategist strains consume primarily glucose while it is present (though GAL-ON cells consume some galactose during that time as well, Appendix Fig S2E). This observation is consistent with previous observations of the GAL network (Escalante-Chong *et al*, 2015). The GAL-ON pure strategist suffers a fitness disadvantage while glucose is still relatively abundant, but outcompetes the GAL-OFF pure strategist when glucose becomes low and galactose remains. Frequency dependence arises because the more GAL-ON cells there are in a population, the more cells must share the galactose, resulting in fewer divisions on galactose and lower fitness for the GAL-ON phenotype (Appendix Fig S2).

We therefore observe negative frequency dependence during batch co-culturing of the GAL-ON and GAL-OFF strains due to exhaustion of resources, similar to the negative frequency dependence that is expected in the simple foraging game described in Fig 1. However, given that this mechanism of negative frequency dependence involves a changing environment, it superficially resembles the changing environments that are present in bet-hedging models. To demonstrate that the negative frequency that we observe does not rely on a changing environment, we therefore co-cultured the GAL-ON and GAL-OFF cells in pseudo-continuous culture. In particular, we diluted the co-culture by a factor of 2 every three hours, thus mimicking a chemostat operating at constant dilution factor. In these experiments, we found strong negative frequency-dependent selection between the GAL-ON and GAL-OFF strains when cocultured in a glucose + galactose environment, but no frequency dependence in a pure glucose environment (Appendix Fig S11). Negative frequency dependence between the GAL-ON and GAL-OFF strains can therefore be observed in either batch culture, in which resources are depleted, or in continuous culture, in which nutrients are continually being supplied.

The simple foraging game predicts that altering the payoffs alters the stable equilibrium during resource depletion. In other words, if resource A increases, then the stable equilibrium should shift toward a larger fraction of the population specializing in that resource. To test for this phenomenon in the GAL network, we replicated the initial competition of our two pure strategists in eight different concentrations of glucose and galactose. More galactose yields a higher equilibrium fraction of GAL-ON cells, while more glucose yields a lower equilibrium fraction of GAL-ON cells (Fig 3). Mutual invasibility is itself dependent on the ratio of sugar concentrations, since in the limit of a low glucose to galactose ratio, we observe that the GAL-ON pure strategist is preferred at all frequencies. The stable equilibrium between our pure strategists therefore shifts as predicted by a negative frequency-dependent game and follows the same general pattern of responsiveness to the environment as the ratio-sensing wild-type GAL network (Appendix Fig S1). This ratio sensing in the wild-type GAL network may allow cells to tune the mixed strategy to track the environmental payoffs in the short term (Escalante-Chong *et al*, 2015) rather than evolving an altered mixed strategy every time the environment changes, though we should note that environmental sensing in itself is perfectly consistent with other phenotypically heterogeneous strategies such as bet hedging.

Another prediction of multi-resource foraging is that a strain adopting a stable mixed strategy cannot be invaded by either pure strategist. Though the two strategies (one pure and one mixed) may still display negative frequency-dependent selection, they are not mutually invasible. Furthermore, as the population frequency of the mixed strategist approaches one, its advantage over any other strategy disappears. In other words, in the limit of a population consisting entirely of mixed strategists, any single invading cell adopting any strategy (pure or mixed) will have the same fitness as the mixed strategist, and the mixed strategist might then be termed *neutrally uninvasible* at high frequencies. By competing the pure strategist strains (GAL-ON/OFF) with a strain containing the wild-type GAL network (mixed strategist), we determined that the mixed strategist is indeed uninvasible by either pure strategist. Additionally, a competition between pure GAL-OFF and the mixed strategist displays the neutral uninvasibility predicted from the game theoretic model (Fig 4A). The wild-type mixed strategy can spread in a population of GAL-OFF cells, but as the wild-type strategy increases in frequency, its relative fitness approaches unity. Moreover, the wild-type mixed strategists are uninvasible by the GAL-ON pure strategist at all frequencies (Fig 4B), though the interaction does not display strong frequency dependence. The lack of strong frequency dependence between this pair suggests that the dynamics of yeast in mixed sugar environments have some subtle deviations from a simple foraging game.

Observing that the pure GAL-ON and GAL-OFF pure strategists are mutually invasible and that the wild-type mixed strategist is uninvasible by either is indicative of the kinds of frequency-dependent interactions that might drive the evolution of phenotypic heterogeneity, but it falls short of providing a direct link between the mutual invasibility of pure strategists and the phenotypic heterogeneity of the wild-type GAL network. This is for two reasons: Firstly, as mentioned above, although the phenotypes may be mutually invasible in the laboratory, it is unlikely that those are the conditions under which the W303 wild-type GAL network actually evolved. Secondly, the pure strategist strains used here are not

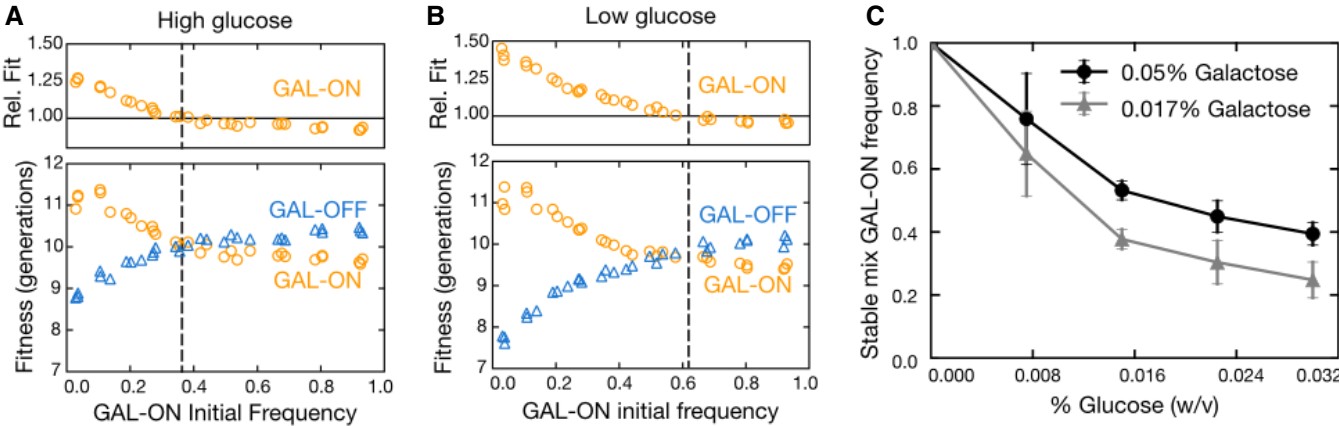

**Figure 3. Altering sugar concentration adjusts phenotypic fitness and equilibrium fractions accordingly.**

A, B    The relative fitness of the GAL-ON pure strategist and absolute fitness (in generations, as in Fig 2D) of both pure strategists are shown for 30 different populations at varying initial frequency of GAL-ON. Data are shown for 0.05% galactose and two conditions: high glucose (0.03%, A) and low glucose (0.017%, B). The GAL-ON pure strategist undergoes roughly the same number of divisions between the two conditions, while the GAL-OFF pure strategists are more fit in higher glucose. Stable equilibrium frequencies (calculated by polynomial spline fitting of relative fitness data) are shown as black dotted lines. Lower glucose results in a higher frequency of GAL-ON cells at the stable mix.

C    Equilibrium GAL-ON pure strategist frequencies as a function of increasing glucose concentrations. Data are shown for high (0.05%, circles) and low (0.017%, triangles) galactose. All equilibria were calculated by polynomial spline fitting of relative fitness data (error bars are 95% confidence intervals; $n = 3$).

Source data are available online for this figure.

perfect approximations of the respective wild-type phenotypes. For instance, the GAL-repressed subpopulation of the wild-type mixed strategist rapidly activates its GAL network upon glucose depletion (Appendix Fig S4), consistent with a classic diauxic growth strategy. However, because of the doxycycline induction of *GAL80*, the GAL-OFF pure strategist takes much longer to transition to GAL-ON when glucose is consumed, effectively suppressing GAL expression within the entire time frame of the competition experiments (Appendix Fig S3). Additionally, the GAL-ON pure strategist's induction activates the GAL network to a greater degree than the

induction in the wild type (Fig 1D), resulting in slightly different costs for expressing the GAL network. This may in fact be the reason that the GAL-ON mutant is less fit than the wild type at all frequencies. For all of these reasons, then, we should not expect quantitative agreement between the fitness dynamics of the OFF and ON pure strategists and those of the wild-type OFF and ON phenotypes.

To more directly investigate the link between multi-resourced environments and phenotypic heterogeneity, we evolved eight replicates of each pure strategist strain over 250 generations in the presence of doxycycline under three resource conditions: pure glucose, pure galactose, and a mixture of glucose and galactose. Cultures were diluted daily 1,000× into fresh media, and GAL activation levels over time were determined via flow cytometry (Fig 5). Starting from the GAL-OFF glucose specialist strain (Fig 5A), all eight populations in mixed sugars (purple) evolved a phenotypic mix of GAL-OFF and GAL-ON. In contrast, a pure galactose (orange) condition drove the rapid evolution of total GAL activation in six of the eight populations (the remaining two replicates were driven to extinction). The eight GAL-OFF populations in pure glucose (blue) remained GAL-OFF throughout. Likewise, starting from the GAL-ON galactose specialist strain (Fig 5B), all eight populations in mixed glucose and galactose similarly evolved a phenotypic mix of GAL-OFF and GAL-ON, while the strain in pure galactose remained essentially GAL-ON throughout. In pure glucose, the GAL-ON strain adopted a very low level of GAL expression, and only a small fraction of the GAL expression typically adopted by GAL-ON subpopulations. This expression remained essentially unchanged throughout the experiment. The fraction of GAL-ON in the pure glucose condition is not plotted in Fig 5, as the unimodal GAL distribution cannot be characterized as GAL-ON or GAL-OFF using the same threshold level of YFP fluorescence used for the other measurements

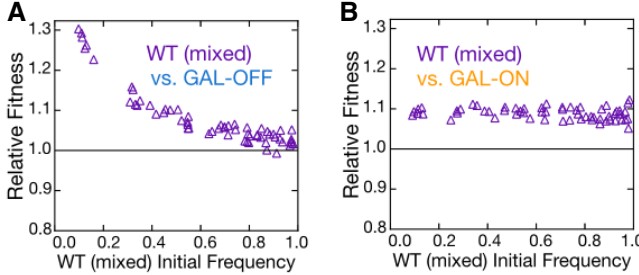

**Figure 4. Wild-type mixed strategist invades both pure strategists and is uninvasible by either.**

The relative fitness of the wild-type mixed strategist over the GAL-OFF pure strategist (A) and GAL-ON pure strategist (B). is shown. Low frequencies of the mixed strategist invade strongly in populations dominated by either pure strategist. As expected of an evolutionarily stable mixed strategy, the relative fitness of the mixed strategist to the GAL-OFF pure strategist approaches one in populations dominated by the mixed strategist. However, the mixed strategist does not display frequency dependence against the GAL-ON pure strategist.

Source data are available online for this figure.

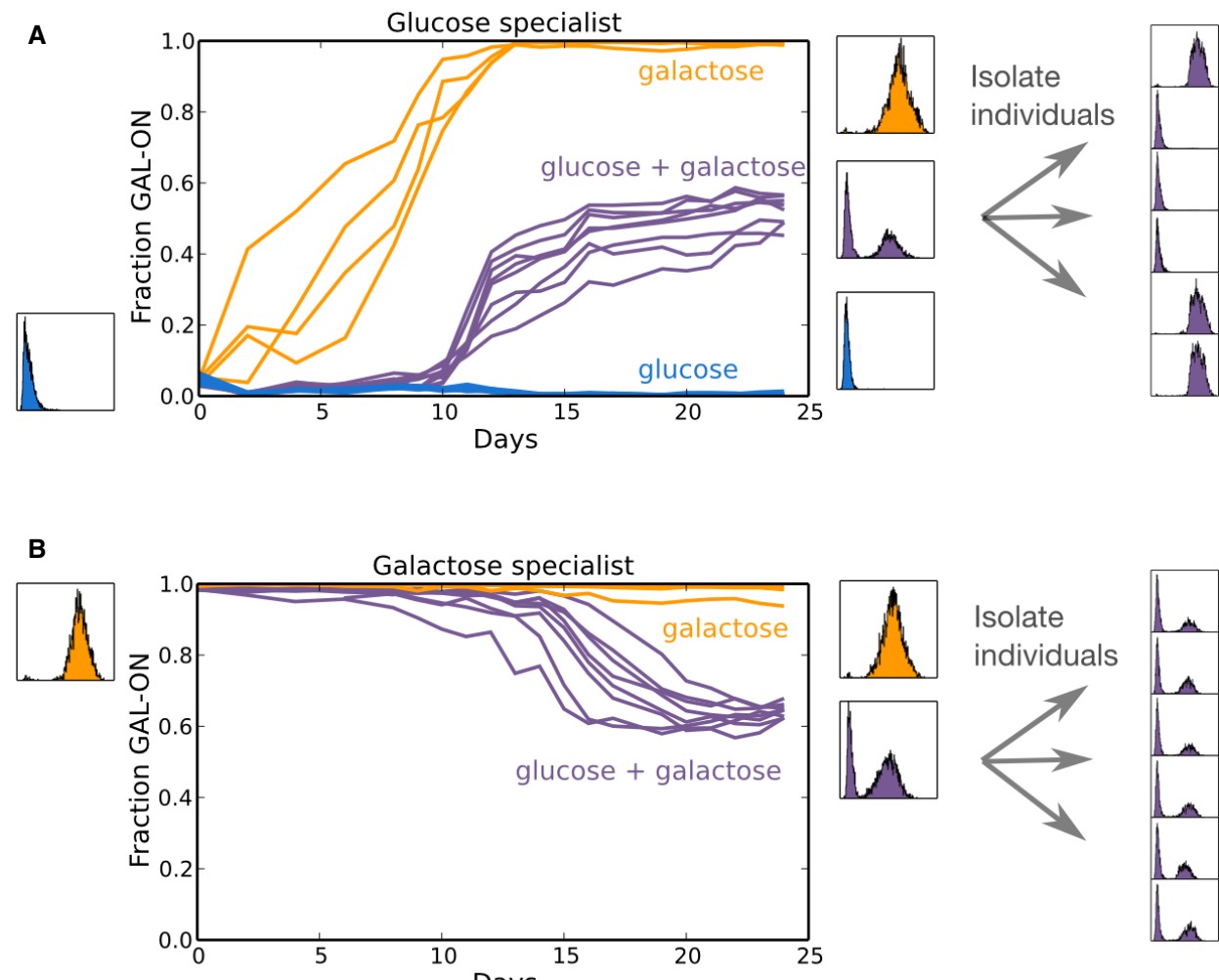

**Figure 5.  Frequency dependence from a mixed resource environment drives the evolution of both phenotypic and genetic heterogeneity.**

A, B   Eight replicates of each of the two specialist strains (GAL-OFF, A, and GAL-ON, B) were incubated in the presence of doxycycline and three separate sugar conditions: 0.1% glucose (blue), 0.1% galactose (orange), and a mixture of 0.03% glucose and 0.05% galactose (purple). Cultures were diluted 1,000× daily into fresh media after reaching saturation. To determine the composition of the evolved mixed populations, cultures were plated on agar and individual colonies were assayed for GAL activation in mixed sugars. (A) Starting from a glucose specialist strain and in the presence of galactose, a mutant pure strategist GAL-ON strain arose spontaneously. In pure galactose, the strain eventually took over the population (orange), while in the mixed resource condition, it evolved toward a stable equilibrium with the GAL-OFF strain (purple, right panel). (B) Starting from a galactose specialist strain in the presence of mixed sugars, the population similarly evolved to a stable mix of GAL-ON and GAL-OFF, but colony purification revealed that the population had evolved to a clonal population of mixed strategists rather than coexistence of pure strategists. See Appendix Figs S8 and S9 for YFP histograms of all mixed sugar replicates. Fraction GAL-ON is not shown for the 1% glucose condition because in that condition, the GAL-ON pure strategist adopts a very low-level unimodal activation state straddling the ON/OFF fluorescence threshold (see Appendix Fig S10).

Source data are available online for this figure.

(Appendix Fig S8). Our laboratory evolution experiments therefore yielded a phenotypic mix of GAL-ON and GAL-OFF only in the environments containing a mixture of glucose and galactose, consistent with our competition experiments between the pure GAL-ON and GAL-OFF strains.

To determine whether the heterogeneous populations that evolved in the mixed resource conditions represented a stable coexistence of different strains or a population of mixed strategists, we isolated individuals through colony purification on agar plates and then incubated them separately in the presence of doxycycline and in mixed glucose and galactose. In all eight replicates evolving from the GAL-OFF strain, we found that pure GAL-ON mutants had arisen and come to a stable coexistence with the pure GAL-OFF strain. In contrast, all eight GAL-ON replicate populations had evolved into primarily clonal populations of mixed strategists similar to the wild-type strain (Fig 5A and B, right panels, see Appendix Figs S8 and S9 for replicate data). Further study will be needed to investigate why the two backgrounds consistently evolved different versions of a similar phenotypic mix (stable coexistence and mixed strategy) in the mixed sugar environment, but the fact that a coexistence of pure

strategies evolved in the same mixed sugar environment that drives evolution of a mixed strategy argues strongly for negative frequency dependence as the driver of phenotypic heterogeneity in this experiment. This is because while both environmental uncertainty and negative frequency dependence are capable of driving the evolution of mixed strategies in clonal populations, only negative frequency-dependent mutual invasibility also drives the stable coexistence of pure strategies. Put differently, bet hedging generally cannot be mimicked by coexistence of strains expressing pure strategies; their evolutionary stability comes exclusively through stochastic adoption of phenotypes. Taken together, then, the results from the evolution of pure GAL-ON and GAL-OFF background directly demonstrate that phenotypic heterogeneity in clonal populations can arise from negative frequency-dependent foraging interactions.

## Discussion

When observing phenotypic heterogeneity in microbial populations, it is important to consider the underlying evolutionary reasons for heterogeneity and distinguish between the different explanations where possible. In this work, we have demonstrated a simple method of determining experimentally whether observed phenotypic heterogeneity is implementing a stable mixed strategy in response to negative frequency-dependent interactions: by isolating the relevant phenotypes and probing them for mutual invasibility based on negative frequency-dependent interactions. We have demonstrated this in the context of the yeast GAL network. In the same context, we have also verified the theoretical prediction that the evolutionarily stable mixed strategy resulting from such frequency dependence is not necessarily optimal for population growth over the course of resource depletion. We have confirmed that the wild-type mixed strategist W303 strain is uninvasible by either pure strategist. We have also evolved a new mixed strategist strain directly from a GAL-ON pure strategist strain by incubating it over many generations in a mixed glucose and galactose environment.

Given that the GAL-ON pure strategist, which is the background strain of the evolved mixed strategist, represents essentially an induced overexpression of *GAL3* relative to wild type, one might expect the evolution of a mixed strategist to be relatively straightforward. However, we also might consider other strategies for coping with multiple sugars to be just as easy, if not much easier, to evolve. These include: loss of function of any one of the functional GAL genes, resulting in an opposite pure strategist GAL-OFF strain that then evolves to a stable coexistence, the evolution of the uniform diauxic strategy commonly observed in mixed resource environments, or, alternatively, the evolution of a "graded response" of low-level GAL activation, a response which the *GAL3*-induced GAL-ON background strain has already been shown to be capable of adopting (Acar *et al*, 2005). Furthermore, the evolved strain did not merely evolve back the wild-type GAL network, since it is not identical to the wild type in its response to glucose and galactose. Specifically, the GAL-inactive subpopulation of the evolved mixed strategist does not activate its GAL network on glucose depletion, as does the wild-type mixed strategist. Rather, it remains GAL-OFF even after glucose is depleted. Further investigation of the molecular mechanisms by which the evolved and wild-type mixed strategist

strains differ will doubtless yield insights into mechanisms of phenotypic heterogeneity in the GAL network, and the evolutionary pathways by which such heterogeneity might evolve.

Appendix Text S1 outlines some guidelines for distinguishing bet-hedging and stable mixed strategies via observation of the fitness dynamics between the represented phenotypes. The definitive characteristic of a stable mixed strategy is that the pure strategists show frequency-dependent mutual invasibility and the nonheritable phenotypic heterogeneity essentially mimics the stable coexistence of heritable pure strategies. All strategies in the population have equal fitness at equilibrium. Indeed, in a population that adopts the stable mixed strategy, any rare mutant with any variant mixed (or pure) strategy is equally fit, whereas in bet hedging some strategies are less fit.

A third class of phenotypic heterogeneity, distinct from both bet-hedging and evolutionarily stable mixed strategies, bears mentioning here: the cooperative "division of labor" strategy, wherein one phenotype sacrifices some of its fitness to improve the fitness of the population as a whole (Ackermann *et al*, 2008; Diard *et al*, 2013). As an extreme example, some colicin-mediated bacterial warfare involves "suicide bomber" phenotypes, wherein release of colicins occurs only with the death of the producing cell (Cascales *et al*, 2007). Cooperative divisions of labor are distinguished by an incentive for individuals to "cheat" by never adopting the low-fitness altruistic phenotype, and so must be maintained through group or kin selection (Nowak, 2006). As in bet hedging, cooperative divisions of labor differ from evolutionarily stable mixed strategies in that they propagate "low-fitness" phenotypes. In bet hedging, these phenotypes are only useful in case the environment changes, and in divisions of labor, they exist primarily to raise the fitness of the other phenotypes and the population as a whole. Bet hedging and divisions of labor can therefore explain phenotypic heterogeneity in a clonal population, but cannot explain heterogeneity resulting from stable genetic coexistence since less-fit genotypes would be competed out of the population over time. In contrast, the mix of phenotypes resulting from negative frequency dependence is stable whether it represents genetic coexistence or clonal phenotypic heterogeneity.

Identifying which of these three evolutionary drivers (negative frequency-dependent interactions, uncertain environment, altruism) is at work in a given phenotypically heterogeneous population is complicated by the possibility that multiple of these phenomena can coexist in a given system. For example, bet-hedging models of phenotypic heterogeneity can, in addition to environmental uncertainty, easily incorporate frequency or density-dependent terms to account for resource limitations. Likewise, models of divisions of labor may also take into account environmental uncertainty. The strategies resulting from multiple drivers might be considered hybrids of the two (Arnoldini *et al*, 2014). In such hybrid models, however, it is important to recognize that although, for instance, environmental uncertainty and negative frequency dependence may exist in the same environment and may both tune the optimal mixed strategy, one or the other force may alone explain the phenomenon of phenotypic heterogeneity, or they both may be required (Rees *et al*, 2010).

In our investigation of heterogeneity in the yeast GAL network, we have shown that frequency-dependent foraging games are logical drivers of heterogeneity in metabolic networks. We have also shown

that such interactions alone can drive evolution of phenotypic heterogeneity from a phenotypically homogenous population. However, we do not rule out the possibility that such heterogeneity may also constitute a bet-hedging or division of labor strategy under appropriate conditions of environmental uncertainty or kin/group selection. In particular, we have argued that because bet-hedging optima are theoretically growth-optimal for a population, the observation of our non-optimal stable mixed strategy is more consistent with the stable point of a negative frequency-dependent game than with the stable point of a bet-hedging scenario (particularly given that clear frequency dependence exists between the phenotypes in a range of environments). Theoretically, a non-growth-optimal bet-hedging strategy would be susceptible to invasion by a mutant that implements a phenotypic mix that provides a higher long-term population growth. The same is not true of a frequency-dependent game (e.g., in the hawk–dove game an isogenic population following the "always-dove" strategy has the highest fitness but such a strategy nonetheless fails to invade the less-fit population composed of the stable mix of hawks and doves). However, we recognize that even a non-optimal bet-hedging strategy is still a bet-hedging strategy if over many environmental shifts the overall population fitness is higher with the phenotypic mix than with either phenotype alone. Furthermore, in complex environmental and metabolic scenarios, there are many reasons for which the optimal bet-hedging strategy may be evolutionarily inaccessible to a population in an uncertain environment. We cannot therefore rule out bet hedging as a possible co-contributor to the observed phenotypic heterogeneity in the yeast GAL network.

Regarding terminology, in this work we examined a case where phenotypic heterogeneity can arise in a clonal population as a result of frequency-dependent interactions and have termed it an "evolutionarily stable mixed strategy" (or mixed ESS). This term originated decades ago when the strategy was described in the context of the hawk–dove model of animal conflict (Maynard Smith, 1982). We recognize, however, that the terms *evolutionary stability* and mixed strategy do not always refer specifically to the case of frequency dependence. An *evolutionarily stable strategy* is merely a strategy that cannot be invaded by an initially rare mutant, while a mixed strategy refers to any probability distribution over pure strategies. Neither of these terms, then, are specific to frequency-dependent selection. However, as de Jong and Kuipers note in their review of the topic, there is no commonly used expression for the stable phenotypically heterogeneous strategy resulting specifically from negative frequency dependence (de Jong *et al*, 2011). In this work, we have opted to use the term *evolutionarily stable* mixed strategy to refer exclusively to the phenotypically heterogeneous strategy resulting from negative frequency dependence.

Given that frequency-dependent interactions are often invoked as reasons for stable genetic coexistence, and evolutionary stable mixed strategies (in the context of hawk–dove games) are central to evolutionary game theory, it is remarkable that this broad class of interactions has received almost no attention as a possible evolutionary reason for phenotypic heterogeneity in clonal microbial populations. To our knowledge, this work constitutes the first experimental evidence that phenotypic diversity in an isogenic microbial population can be driven by such interactions in multi-resource foraging games. It remains to be seen exactly to what degree negative frequency dependence is responsible for the widespread phenotypic heterogeneity in isogenic populations.

# Materials and Methods

### Strains

The three strains of *Saccharomyces cerevisiae* (wild-type mixed strategist, GAL-OFF specialist, and GAL-ON specialist) are modified from those used in Acar *et al* (2005), which were derived from the diploid W303 strain of *S. cerevisiae*. All strains have a $ADE2$-$P_{GAL1}$-$YFP$ reporter construct inserted at one $ade2$ site for monitoring activation of the GAL network. Since one $ura3$ locus was already occupied by inducible forms of $GAL80$ or $GAL3$, yeast was first sporulated to isolate the remaining $ura3$ locus. Identity of the haploids was confirmed by replica plating. Haploids containing $ura3$ were then transformed with the yeast integrating vector pRS306 containing URA3 and either RFP(tdTomato) or CFP cloned downstream of a TEF1 promoter. Constitutive fluorescence was confirmed by microscopy and flow cytometry. Fluorescent cells were then mated with the appropriate haploid to produce the desired strain. All strains were maintained on synthetic media his- and ura-agar dropout plates supplemented with 2% glucose.

The Gal80-inducible (GAL-OFF pure strategist) strain has a double $GAL80$ deletion. $P_{TETO2}$-$GAL80$ is inserted at one $ura3$ locus, while $P_{MYO2}$-$rtTA$ is inserted at an $ade2$ locus. The GAL3-inducible (GAL-ON pure strategist) strain has a double $GAL3$ deletion with $P_{TETO2}$-$GAL3$ inserted at a $ura3$ locus and $P_{MYO2}$-$rtTA$ inserted at an $ade2$ locus. Complete genotypes for the strains are found in Appendix Table S1.

### Competitions

To initiate doxycycline induction in pure strategists, strains were initially mixed at desired initial frequencies from plated colonies, then incubated in 1.0 µg/ml doxycycline and 0.01% (w/v) glucose for 24 h from a starting density of ~3 × 10^4 cells/ml to a saturating density of ~6 × 10^6 cells/ml, and then diluted to ~3 × 10^4 cells/ml in synthetic media supplemented with glucose and galactose as indicated. Fractions were measured before and after incubation using a Miltenyi MACSquant flow cytometer (20,000+ cells per well), and population density was measured as absorbance at 600 nm in a microplate spectrophotometer (conversions assume 3 × 10^7 cells/ml at $A_{600}$ = 1.0). Fitness of strains was calculated as the number of generations of growth undergone during the incubation. Absolute fitness for each strain was calculated as the number of doublings:

$$W_{abs} = \log_2\left[\frac{OD_f * f_f}{OD_i * f_i}\right]$$

Relative fitness for each strain was calculated as follows:

$$W_{rel} = \ln\left[\frac{OD_f * f_f}{OD_i * f_i}\right] \Big/ \ln\left[\frac{OD_f * (1 - f_f)}{OD_i * (1 - f_i)}\right]$$

where $f_i$ and $f_f$ are the initial and final fractions of the strain in the population, and $OD_i$ and $OD_f$ are the initial and final population

densities, respectively. Competitions involving the wild-type GAL network were necessarily limited to a single day because the wild-type GAL-OFF fraction switches to GAL-ON when all the galactose is consumed. Subsequent days, beginning from the GAL-ON history, do not adopt a bimodal state in mixed sugars. Rather, the population adopts a very wide intermediate distribution of GAL activation which remains essentially unchanged over the course of multiple rounds of growth in mixed sugars.

**Evolution of mixed strategists from pure strategists**

Pure strategist Gal80-inducible (Gal80i or GAL-OFF) and Gal3-inducible (Gal3i or GAL-ON) strains were initially doxycycline-induced in media containing YNB, CSM, 1 µg/ml doxycycline, and 0.01% glucose for 24 h to saturation (~OD 0.25). Then they were diluted 100-fold into 0.1% glucose, 0.1% galactose, and a mixture of 0.03% glu/0.05% gal. After consuming all the sugars and reaching saturation, they were diluted 1,000× into fresh media and allowed to resume growing. This process was repeated on a daily basis for 26 days; however, due to two early rounds of 200× and 500× dilution, respectively, the total number of generations of growth over the 26 days was ~250. GAL activation states were analyzed via flow cytometry. To explore the composition of the evolved populations, 30 µl of evolved saturated culture was streaked on agar plates containing his and ura dropout YNB/CSM and 2% glucose. After 2 days of growth, individual colonies were suspended in separate wells of a 96-well plate and grown for 2 additional cycles in the presence of doxycycline and the same sugar mix they evolved in, and the resulting populations' GAL activation states were measured via flow cytometry. All evolution experiments were performed in flat-bottomed 96-well culture plates incubated at 30°C.

**Expanded View** for this article is available online.

## Acknowledgements
The authors would like to thank D. Kessler, A. Sanchez, A. Velenich, L. Gibson, T. Gillespie, M. Springer, M. Laub, and M. Vander Heiden, for helpful discussions, and the members of the Gore Laboratory for helpful comments on the manuscript. M. Acar provided the parent diploid strains from which present strains were modified. This work was supported by the Allen Distinguished Investigator Program and the NIH New Innovator Award. JG also acknowledges the support of as a Pew Scholar in the Biomedical Sciences and a Sloan Research Fellow.

## Author contributions
DH and JG designed the study and performed analysis. DH and KA performed the experiments. DH and JG wrote the manuscript.

## Conflict of interest
The authors declare that they have no conflict of interest.

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
