## [Review Process File · Molecular Systems Biology]

Negative frequency dependent interactions can underlie phenotypic heterogeneity in a clonal microbial population

David Healey, Kevin Axelrod and Jeff Gore

Corresponding author: Jeff Gore, MIT

Review timeline:

First submission:	18 June 2015
Editorial Decision:	31 July 2015
Revision received:	03 November 2015
Editorial Decision:	10 December 2015
Second submission:	25 April 2016
Editorial Decision:	14 June 2016
Revision received:	28 June 2016
Accepted:	04 July 2016

Editor: Maria Polychronidou

Transaction Report:

1st Editorial Decision

31 July 2015

Thank you again for submitting your work to Molecular Systems Biology. We have now heard back from the three referees who agreed to evaluate your manuscript. As you will see from the reports below, the referees think that the presented findings seem interesting. However, they raise a series of concerns, which should be carefully addressed in a revision of the manuscript.

The referees' recommendations are rather clear so there is no need to repeat all the points listed below. In sum, reviewer #3 recommends performing further analyses in order to better support the presented conclusions. Moreover, all reviewers raise some issues related to the discussion and presentation of the data and the main findings. In line with the comments of reviewer #1, we think that information (both text and figures) that is essential for understanding the main conclusions of the study should be moved in the main text.

REFeree REPORTS

Reviewer #1:

In their manuscript "Negative frequency dependent interactions can underlie phenotypic heterogeneity in a clonal microbial population", Healy and coauthors show that phenotypic heterogeneity of a population can be evolutionary stable in homogeneous environments, even when bet hedging and division of labor are not at work. They argue that the observed heterogeneity in

their model population (yeast thriving on glucose and, by stochastically activating the GAL network, galactose) is due to negative frequency dependent interactions between GAL-ON and GAL-OFF phenotypes, which are a result of the multi-resource environment they live in.

Overall, the manuscript presents convincing arguments for the above-mentioned hypotheses. The research subject is very interesting due to its widespread applicability and conceptual importance for understanding the emergence of phenotypic heterogeneity. The experimental evidence is mostly solid and sufficient to support the claims of the manuscript (except for the questions below). In addition, plenty of theoretical background is provided in the supplemental material that strengthens the key arguments made to differentiate the different scenarios in which phenotypic heterogeneity might emerge. However, we think that the manuscript could be significantly improved by being more precise in setting up the model system, adopting a more bottom-up approach in presenting the arguments and also by moving some important statements from the supplementary material to the main text. See our specific concerns below.

General presentation issues:

1. In motivating their work, authors appeal to the classical ecological models demonstrating "negative frequency-dependent selection." However, one must be careful in drawing such parallels since the effect studied in the paper is rather different from the negative frequency-dependent selection in standard ecological models, such as the classical Hawk-Dove model. In the latter, the environment is kept fixed, and the negative frequency selection occurs due to direct interaction between different phenotypes that affects the cost of foraging. In the system under study, however, the direct interaction is absent, and the negative selection effect only is observed after the resources have been strongly depleted. This poses some concern in regards to using the established term "Negative frequency-dependent selection" in this rather different scenario.

At the very least, this distinction should have been made very clear from the beginning, and it was not. The authors were not very thorough in specifying the experimental conditions, and for the first few pages of the manuscript, the reader could be under the impression that the study is done in a chemostat-type setting with constant sugar supply to the cells, whereas in reality the indicated concentrations of glucose and galactose are only initial values, and they change over the course of experiments while resources are being depleted. This in turn leads to a rather non-trivial time dependence of the fitness of GAL-ON and GAL-OFF phenotypes in this non-stationary experimental environment. Furthermore, it appears that the outcome of the experiment should also non-trivially depend on the volume of the reactor (which is not mentioned at all), and in a bigger reactor one would have to wait much longer to see the negative frequency-dependent selection, or the effect may even be completely masked by other factors. We do think the overall effect is real, but it has to be made very clear that it only occurs when the environment fluctuates from rich to poor and back. We think that the negative frequency-dependent selection framework only applies at a certain level of abstraction that is not discussed at all, namely if one considers discrete time steps instead of continuous time and these time steps are tied to the resource depletion cycles (akin to a Poincaré map for a continuous time dynamical system). We believe these issues have to be explicitly discussed.

We also note in this regard that the required time-dependent environment makes the usage of the term "unique stable equilibrium" somewhat questionable. The authors use this term to emphasize that the relative frequency of GAL-ON phenotypes is the same before and after the 20-hr experiment during which the resources have been depleted. But during the experiment, as seen in Figure S2B, the frequency of GAL-ON undergoes significant transient changes before returning to the initial value. This behavior is more akin a fixed point of a Poincaré map that relates the frequency of GAL-ON before the growth cycle to the one after it. The term "stable equilibrium" implies that the frequencies of GAL-ON and GAL-OFF are not changing in time during the experiment, which is not the case here.

2. We feel that the authors should present the acquired data and the involved concepts in a more bottom-up manner in order to allow the reader to understand the arguments when first presented. Some important thoughts in the introductory section of the "Results" section have been completely "outsourced" into the Supplement text and/or supplementary figures.

For example, on line 108, Figure S2 is said to show a significant metabolic cost for expressing GAL, but this is not explained any further. Also, in addition to explaining why the figure shows that, the text should refer to the panel in which the evidence can be found (S2D?). Within figure S2, the reader is confronted with "the 20-hr competition between GAL-ON and GAL-OFF pure strategists", fitness measures, etc, things that have not been introduced in the main text and will not be for a while. Also, in the main text, the order of the first figures that are mentioned is S1, S4, S2 (and before any figures in the main text), although they present important results that set the stage for the conclusions of the paper.

Another example is the end of the Introduction (line 79), where the discussion and Table 1 are referenced although the arguments that are necessary to understand the "check marks and crosses" in Table 1 have not been presented. In fact, the most convincing arguments for these are hidden in the Supporting text 1 and are never mentioned in the main text, neither there, nor later. At least some of this discussion needs to be in the main text, so it would help readers understand why naive considerations that could lead to different "check marks and crosses" in Table 1 are wrong. Specifically, on page 9 of the supplement (second and third paragraph), there is a very convincing discussion about why not all strategies are equal in bet-hedging, even if one considers very long times with a mixture of environments. Some condensed version of this should go to the main text. Furthermore, the division of labor is only mentioned in passing even in the Supporting text. It would help the reader and make the arguments and Table 1 much stronger there was a more specific explanation of the division of labor in the context of this study.

More specific comments:

- Figure 2E: Why are the population densities plotted at 16h as opposed to end of the 20h competition experiment like in all other data? This unnecessarily weakens the point the authors are trying to make: that the evolutionary stable state is not one that maximizes population growth. Because of the non-trivial population dynamics that is happening at the time when glucose gets depleted (e.g., GAL-ON frequency first decreasing and then increasing in figure S2B), it is essential to always compare the same time points, so all possible effects are always included, and there is no danger of observing a transient state (for example, in figure 2E, the population around .5 GAL-ON might grow more in the last four hours, rendering it optimal). If the population sizes at 20h are already too saturated for lower GAL-ON frequencies, maybe it would be advisable to plot *both* time points in different colors. That way, it would become clear that lower GAL-ON frequencies saturate later and there is no late "picking up" of population growth.

- To prove mutual invasibility, competition experiments between pure GAL-ON and GAL-OFF strategists were evaluated. Figure 2C shows that the fraction of GAL-ON and OFF strategists increases over time when initially rare. However, Figure S2B shows data from similar (or the same?) experiments. Here, a low initial GAL-ON frequency increases towards the end of the experiment, consistent with Figure 2C, but for high initial GAL-ON frequencies (i.e. rare GAL-OFF), the curves don't show a clear trend toward lower GAL-ON frequencies (i.e. more GAL-OFF) toward the end. According to Fig 2C, the downward shift in GAL-ON frequency is relatively small for this case (and also the difference in fitness according to Fig 2D), so maybe it is just a matter of scaling. The reader shouldn't be left alone to find out how these two plots can be reconciled.

- The caption of figure S2 says that colors indicate initial frequencies of the GAL-ON pure strategist. Most certainly, Figure S2D does not contain data for 0% and 100%, as GAL-ON relative fitness cannot be calculated in this case. The GAL-ON percentages of the lightest curves in Figure S2D should be stated clearly so it is possible to read off the initial GAL-ON frequency that leads to a relative fitness of 1 (after 20h) from the graph. Also, in figure S2C, the 0% curve is probably covered by the black curve, so the same thing applies to this panel.

- Starting on line 210 of the main text, GAL-OFF pure strategists are said to take long to transition to GAL-ON after glucose depletion, referencing figure S3 as evidence. However, in figure S3, only wild-type dynamics is shown. We couldn't find this evidence in any other figure.

- When referring to Figure S1 on line 100 of the main text, it is not clear that the observed bimodal distribution of GAL is not just a transient state that is observed because some cells are a little bit

faster switching to GAL-ON than others. Only together with figure S3 (and the plateau phase of GAL-ON in the intermediate galactose case in panel B), it becomes clear that this distribution is not just the snapshot of a transition process from GAL-OFF to ON. It would be optimal to either show two different time points for each plot in Figure S1 or to show that the bimodal activation even persists in an environment with *constant* intermediate levels mixed sugars.

Minor issues:

- Line 193 and 196 of the main text refer to Figure 5, but only Figure 4 makes sense here.
- The line plots in Figure 2C are misleading because they hide the non-trivial time course of the GAL-ON fraction in the population that is shown in the more detailed Figure S2B. It looks like the fractions are steadily increasing/decreasing/constant, which is not actually the case. An alternative way of showing only beginning and end state could be to plot end state against initial frequency and draw an identity line for reference. At least, one should find a way to remove the lines that give a false impression of the time course.
- Line 121 says that the evolutionary stable mix is uninvasible by any phenotypic variant. At this point, it is not clear what "variants" are included here. Clearly, it cannot be any arbitrary phenotype. From the supplement, it seems the authors refer to either pure phenotypes or mixed strategies with different probabilities, but this should be explicitly mentioned.
- It is difficult to differentiate the darkest colors in Figure S2A.
- The axis labels on the little plots in figure S1 are not readable. As they are probably arbitrary units, they can be removed and the figure caption could mention that the axis scaling of all the plots is identical.
- The figure caption for panel S3D is missing.
- The figure caption of figure S6 refers to figure 6. But there is no figure 6 in the main text.
- The caption title (bold text) of figure S7 is identical to that of figure S6 and should be edited to reflect the actual facts.
- In the last line on page 3 of the Supplement, the word "fitness" is missing within the first brackets.
- On page 5 of the Supplement, the enumeration appears to be off by 1.
- One page 9 of the Supplement, the second paragraph ends with an incomplete sentence.
- The last sentence on page 6 of the Supplement refers to Fig. 1. Shouldn't this be Fig 2? Also, the text says $V=2/3 C$, but the figure caption says $C=3V$.

Reviewer #2:

This paper documents how feeding on glucose versus galactose in yeast generates negative frequency dependent selection that leads to a stable equilibrium. Starting from yeast that can feed on both glucose and galactose, the authors generated otherwise-isogenic strains that could grow only on glucose or only on galactose. Mixed at different frequencies, these showed all the hallmarks of negative frequency dependence and did not fit other explanations for the maintenance of two phenotypes. The paper is extremely thorough and includes theoretical models of how feeding on two foods can generate negative frequency dependence. It links, for perhaps the first time, frequency dependent selection with stochastic phenotype switching.

I have not major criticisms. I was a bit puzzled by the introduction, which takes as its starting point stochastic gene expression, which sometimes leads to coexistence of phenotypes. Frequency dependent selection comes in only later. I assume this is because it is the path that the authors own

thinking followed. But I think most evolutionary biologists, when faced with the problem of stability of multiple phenotypes, would think of frequency dependent selection first and stochastic bet hedging somewhere down the list. Perhaps this is a difference in the microbial field, but even there, I think the authors underestimate the amount of work that has addressed frequency dependent selection and more of that work should be referenced (even if it is not related to stochastic expression).

This is not to take anything away from the substance of this paper. Though there are other papers on frequency dependence in microbes, this one is exceptionally rigorous and thorough. And it does make a link to stochastic gene expression that other papers do not make.

Section S1 of the supplement seems unnecessary to me as it mostly (entirely?) re-derives established results. It might be useful as a primer for those who are unfamiliar with bet-hedging or negative frequency dependence, but if it is kept for that reason, it needs to be referenced much more thoroughly.

Section S2 is more specific and more novel and it generates results important for understanding the kind of two-resource frequency dependence studied in the main paper.

Line 106. Should this be Figure S3 instead of S4?

Line 250. Figure 5 instead of Figure 6?

Reviewer #3:

The goal of this manuscript is to test the hypothesis that negative frequency-dependent selection underlies phenotypic heterogeneity in microbial populations. This is examined in the yeast GAL network that is expressed in a subpopulation of yeast cells when they grow in a mixture of glucose and galactose. This is a "mixed strategy" that is evolutionarily stable to "pure strategist" invaders that take up only glucose or only galactose. The manuscript starts by listing a set of criteria that distinguish negative frequency-dependent interactions from two other causes of phenotypic heterogeneity: bet-hedging and division of labor. Experiments are performed to check if the yeast GAL system fulfills some of these criteria. Finally, evolution experiments are performed to observe the conversion of pure strategists into mixed strategists, according to the expectation that once they arise, mixed strategists should invade pure strategists.

The manuscript poses an interesting, novel idea, and the experiments generate interesting data. It is not entirely clear, however, if the data fully support the claim that the yeast GAL network is a clear example of phenotypic heterogeneity arising from negative frequency-dependent selection. It is unclear if the other strategies can be excluded. The manuscript may be publishable if most of the following comments are addressed in a revised version.

(1) The GAL gene expression distributions are recorded at a certain time during a 20-hour time course. However, during these 20 hours the cells run out of glucose, and even most of the galactose in the growth medium. Therefore, the glucose/galactose ratio will be entirely different at the end of the time course compared to the beginning. Consequently, the GAL gene expression distribution (the fractions of On and Off cells) will change as well such that in the end all cells may be GAL-On (as if they were pure strategists). These distributions should be measured during the 20-hour time course to determine how they change. How does this affect the model and the conclusions?

(2) One underlying assumption is that cells do not switch from one subpopulation to the other. This is based on Fig. S4. However, Fig. S4 does not support this claim. In fact, cells sorted from the Off subpopulation seem to switch very quickly On (in a matter of hours). The "memory" of similar transitions has been measured before in yeast, and it can be up to days or even weeks (see PMID15889097 and PMID18362885). For example, in PMID18362885, switching is "slow" if it happens in ~100 hours or more, and even "fast" switching happens over more than 15 hours. Compared to these values, the switching rates in the GAL system actually seem super-fast. They should be measured exactly, in both directions (On to Off and Off to On) and incorporated in computational models to determine their effect. The rates should be determined based on the

"Population dynamics model" in PMID18362885, to account for different growth rates of GAL-On and GAL-Off cells.

(3) Mutual invasion of pure strategists: this probably depends on the initial % glucose and % galactose in the medium. That is, if there is ~ 1 or 2 orders of magnitude more galactose than glucose then GAL-Off purists will not reach high enough numbers at the end of the time course to be considered invading (simply because they do not have enough sugar to grow on). The reverse may also be true. This should be investigated more carefully and the manuscript should be revised accordingly.

(4) Fig. S2 has no error bars on any of the panels. Were these values measured in triplicate? If not, why not? If yes, are the error bars smaller than the symbols? This should be clearly stated. How exactly is Fig. 2C obtained from Fig. S2? This should be very clearly explained.

(5) The models presented in the S1 Text are very simplistic game-theoretic models. It would be helpful to try other models, especially for bet-hedging. For example, the differential equation model in PMID18362885 (or its stochastic implementation) would be very useful to try. The conclusions should be robust to the type of model used. Moreover, more experimental details should be incorporated into models. At least some of the models should be more than simplistic illustrations of ideas: they should be rooted in and should be constrained by experimental observations.

(6) There was no sequencing performed to confirm whether or not the phenotypic mix of GAL-ON and GAL-OFF observed in the evolution experiments (e.g., the mixed "clonal" populations presented in Fig. 5) is the result of a genetic or epigenetic/nongenetic phenomenon. If the changes are in fact nongenetic, then it is debatable whether or not evolution actually occurred. If there are mutations occurring, then which type, where in the GAL network/genome, and if multiple mutations occurred in what order did they occur and is this important? This being the case, further experimentation/sequencing should be performed to elucidate the underlying mechanisms of phenotypic variability in the GAL network.

(7) One interesting experiment to run is to compete a mixture of the "pure" strategists with the mixed strategist. That is, a 3-strain competition. Would the mixed strategist invade the mixture of the pure strategists or vice versa?

Minor comments:

(1) It was mentioned that the study is "in the context of an exponentially growing microbial population", however cell populations were allowed to reach saturation? Why were the cells not maintained in exponential growth phase by using more frequent dilutions of larger cell culture volumes? At saturation, starvation and other responses in budding yeast may be activated which makes for a less controlled experiment.

(2) Apparently, the peak positions shift in Fig. S1. How would this affect the models and the conclusions?

(3) It was not clear immediately why a higher growth penalty for phenotype B results in a higher phenotype B fraction in the growth optimal mix? Perhaps the authors can provide a more intuitive explanation for the reader.

(4) Top of Page 5: I think the text tries to refer to Fig. 4.

(5) The manuscript would benefit from the elimination of some jargon and game-theory terminology.

Reviewer #1:

In their manuscript "Negative frequency dependent interactions can underlie phenotypic heterogeneity in a clonal microbial population", Healy and coauthors show that phenotypic heterogeneity of a population can be evolutionary stable in homogeneous environments, even when bet hedging and division of labor are not at work. They argue that the observed heterogeneity in their model population (yeast thriving on glucose and, by stochastically activating the GAL network, galactose) is due to negative frequency dependent interactions between GAL-ON and GAL-OFF phenotypes, which are a result of the multi-resource environment they live in.

Overall, the manuscript presents convincing arguments for the above-mentioned hypotheses. The research subject is very interesting due to its widespread applicability and conceptual importance for understanding the emergence of phenotypic heterogeneity. The experimental evidence is mostly solid and sufficient to support the claims of the manuscript (except for the questions below). In addition, plenty of theoretical background is provided in the supplemental material that strengthens the key arguments made to differentiate the different scenarios in which phenotypic heterogeneity might emerge. However, we think that the manuscript could be significantly improved by being more precise in setting up the model system, adopting a more bottom-up approach in presenting the arguments and also by moving some important statements from the supplementary material to the main text. See our specific concerns below.

We thank reviewer #1 for their overall positive assessment of the study. Below we respond in detail regarding how we have addressed the concerns.

General presentation issues:

1. In motivating their work, authors appeal to the classical ecological models demonstrating "negative frequency-dependent selection." However, one must be careful in drawing such parallels since the effect studied in the paper is rather different from the negative frequencydependent selection in standard ecological models, such as the classical Hawk-Dove model. In the latter, the environment is kept fixed, and the negative frequency selection occurs due to direct interaction between different phenotypes that affects the cost of foraging. In the system under study, however, the direct interaction is absent, and the negative selection effect only is observed after the resources have been strongly depleted. This poses some concern in regards to using the established term "Negative frequency-dependent selection" in this rather different scenario.

We agree that the experimental system that we explore here is more complicated than the simple game-theory models such as the hawk-dove game. However, we note that even in the simple animal foraging game, the interaction is mediated through resource availability and depletion, so in this sense it is similar to the indirect interactions that we study experimentally in the galactose / glucose system. Given this, we believe that "negative frequency dependent selection" remains an appropriate description of our experimental system.

At the very least, this distinction should have been made very clear from the beginning, and it was not. The authors were not very thorough in specifying the experimental conditions, and for the first few pages of the manuscript, the reader could be under the impression that the study is done in a chemostat-type setting with constant sugar supply to the cells, whereas in reality the indicated concentrations of glucose and galactose are only initial values, and they change over the course of experiments while resources are being depleted. This in turn leads to a rather nontrivial time dependence of the fitness of GAL-ON and GAL-OFF phenotypes in this nonstationary

experimental environment. Furthermore, it appears that the outcome of the experiment should also non-trivially depend on the volume of the reactor (which is not mentioned at all), and in a bigger reactor one would have to wait much longer to see the negative frequency-dependent selection, or the effect may even be completely masked by other factors. We do think the overall effect is real, but it has to be made very clear that it only occurs when the environment fluctuates from rich to poor and back. We think that the negative frequency-dependent selection framework only applies at a certain level of abstraction that is not discussed at all, namely if one considers discrete time steps instead of continuous time and these time steps are tied to the resource depletion cycles (akin to a Poincaré map for a continuous time dynamical system). We believe these issues have to be explicitly discussed.

We apologize that the text was not sufficiently clear regarding this point. To address this, at line 199 we have added the italicized to the following sentence: “To test for negative frequency dependence between the pure strategists, we mixed six biological replicate pairs of RFP-labeled GAL-ON and CFP-labeled GAL-OFF strains at a total of 60 different initial frequencies, and incubated them for in a mixed glucose and galactose environment 0.03% (w/v) glucose and 0.05% galactose for 20 hours, *until sugars were consumed*. (Figure 2C-E)”

Additionally, we have framed the simple foraging game described by Figure 1 as a foraging game “that exists when resources are limited” (line 147), and the negative frequency dependence in that game arises because “The more individuals that adopt the strategy “specialize in A,” *the more quickly A will be consumed*, reducing payout to individuals who choose that strategy.” (line 150) The frequency dependence of the ON and OFF strategies are subsequently framed in the context of that basic scenario, with resource depletion over time.

We note that in principle the volume of the reaction vessel should not influence the results, since it is only concentrations of cells and sugars that enter into the model (doubling the volume does not change the number of sugar molecules available per cell).

We also note in this regard that the required time-dependent environment makes the usage of the term “unique stable equilibrium” somewhat questionable. The authors use this term to emphasize that the relative frequency of GAL-ON phenotypes is the same before and after the 20-hr experiment during which the resources have been depleted. But during the experiment, as seen in Figure S2B, the frequency of GAL-ON undergoes significant transient changes before returning to the initial value. This behavior is more akin a fixed point of a Poincaré map that relates the frequency of GAL-ON before the growth cycle to the one after it. The term “stable equilibrium” implies that the frequencies of GAL-ON and GAL-OFF are not changing in time during the experiment, which is not the case here.

The reviewer #1 has accurately described the dynamics over the course of the day. We are treating the entire 20hr growth cycle as a game, which in principle can be treated as a black box with an input fraction and an output fraction. As the reviewer indicates, the equilibrium fraction is when these fractions are the same. We have of course measured the temporal dynamics of the sugars and strains (Supp Fig S2), but this was intended merely to provide mechanistic insight. In our view it is appropriate to refer to this as a stable equilibrium fraction given the mapping from the fraction at the beginning of the day to the end of the day. In previous work (Yurtsev et al, MSB (2013)) we have used this approach and demonstrated that the fixed point of the single-day difference equation also yields the equilibrium over multiple growth-dilution cycles. However, to avoid confusion on this point, we have made an effort to be more explicit in the main text and the figure caption about the time course during the day.

2. We feel that the authors should present the acquired data and the involved concepts in a more bottom-up manner in order to allow the reader to understand the arguments when first presented. Some important thoughts in the introductory section of the “Results” section have been completely “outsourced” into the Supplement text and/or supplementary figures.

For example, on line 108, Figure S2 is said to show a significant metabolic cost for expressing GAL, but this is not explained any further. Also, in addition to explaining why the figure shows that, the text should refer to the panel in which the evidence can be found (S2D?). Within figure S2, the reader is confronted with “the 20-hr competition between GAL-ON and GAL-OFF pure

strategists", fitness measures, etc, things that have not been introduced in the main text and will not be for a while. Also, in the main text, the order of the first figures that are mentioned is S1, S4, S2 (and before any figures in the main text), although they present important results that set the stage for the conclusions of the paper.

We felt that it would impede the flow and logic of the paper to dwell on this too much in the main text, but we agree that Fig S2 needs interpreting to understand why it relates to the metabolic cost of expressing GAL. To address this, we have added the following sentence in the figure legend for Fig S2: “**B,C,D** With the exception of the 0% and 100% GAL-ON populations, the fraction of GAL-ON initially decreases relative to GAL-OFF (with around a 10% relative fitness cost, **D**), until around the time glucose is depleted (**E**), at which point GAL-ON becomes more fit than GAL-OFF and increases in fraction. Final relative fitness of GAL-ON depends on initial fraction (**D**); low initial fractions of GAL-ON end at higher final fraction, while higher initial fractions of GAL-ON end at lower final fraction”

We also agree that the results from supplementary figures S1 and S4 (which describe the bimodal GAL response to glucose and galactose) are an important “bottom-up” background to understanding the GAL response. These figures were not included in the main text, however, because their results are not novel to the field or intended to be a core finding of the paper, since bimodality and fitness of the GAL network in glucose and galactose has previously characterized. We do provide and reference the figures in the supplementary text to show details of our specific strains, to show accordance with previously published results, and to--as the referee suggests--provide some background and intuition about the GAL response in multiple sugar environments before launching into figures about mutual invasibility and frequency dependence, which are the core findings of the paper.

We recognize that we did not make the purpose of these supplementary figure references sufficiently clear in the main text. We have accordingly made some changes. To clarify that, for instance, the assertion that “activation of the GAL network imposes a metabolic cost on the cell” is neither novel nor intended to be presented as a core finding of the paper, we have moved the reference to supplementary figure S2 after the relevant citations, and referred to S2D only parenthetically as a source of details of the dynamics in the present specific context (Line 144). We have also highlighted the previously published results that pertain to figure S1 both in the main text and in the figure caption, as described below.

Another example is the end of the Introduction (line 79), where the discussion and Table 1 are referenced although the arguments that are necessary to understand the "check marks and crosses" in Table 1 have not been presented. In fact, the most convincing arguments for these are hidden in the Supporting text 1 and are never mentioned in the main text, neither there, nor later. At least some of this discussion needs to be in the main text, so it would help readers understand why naive considerations that could lead to different "check marks and crosses" in Table 1 are wrong. Specifically, on page 9 of the supplement (second and third paragraph), there is a very convincing discussion about why not all strategies are equal in bet-hedging, even if one considers very long times with a mixture of environments. Some condensed version of this should go to the main text. Furthermore, the division of labor is only mentioned in passing even in the Supporting text. It would help the reader and make the arguments and Table 1 much stronger there was a more specific explanation of the division of labor in the context of this study.

We are glad that reviewer #1 felt that our description of bet-hedging was helpful. We put this description in the supplemental section because our paper is intended to be about negative frequency dependent selection rather than bet hedging. However, we acknowledge that both of these ideas are subtle and a proper understanding of each cannot be obtained without thinking about how they are different. For this reason, we have now put a condensed version of Supplementary Discussion into the main text as a standalone box. We hope that this helps.

More specific comments:

- Figure 2E: Why are the population densities plotted at 16h as opposed to end of the 20h competition experiment like in all other data? This unnecessarily weakens the point the authors are trying to make: that the evolutionary stable state is not one that maximizes population

*growth. Because of the non-trivial population dynamics that is happening at the time when glucose gets depleted (e.g., GAL-ON frequency first decreasing and then increasing in figure S2B), it is essential to always compare the same time points, so all possible effects are always included, and there is no danger of observing a transient state (for example, in figure 2E, the population around .5 GAL-ON might grow more in the last four hours, rendering it optimal). If the population sizes at 20h are already too saturated for lower GAL-ON frequencies, maybe it would be advisable to plot *both* time points in different colors. That way, it would become clear that lower GAL-ON frequencies saturate later and there is no late "picking up" of population growth.*

We plotted OD at 16hr in an attempt to provide the simplest possible data to illustrate that the equilibrium fraction does not saturate first. Eventually all the populations saturate, and reviewer #1 surmises correctly that by 20 hours the high and intermediate GAL-ON fraction populations have already saturated too much to show clear growth differences between them. Indeed, since resource depletion is the underlying cause for mutual invasibility in our system, it is fundamentally difficult to show mutual invasibility at the same time point at which it is possible to still see growth differences. To clarify this point, we have added the following to the caption for Figure 2: *"growth at 16 rather than 20 hours is shown because by 20 hours the high and intermediate GAL-ON populations have saturated too much to show growth differences between them"*

However, after considering the argument of reviewer #1 we agree that showing both unsaturated and saturated time points may provide a better sense of the dynamics of the co-cultures. Accordingly, we have added Appendix Figure S7, which depicts both 16hrs and 24 hours (full saturation) in different colors. We note that the very tail end (very low initial GAL-ON) takes around 24 hours rather than 20 to saturate. To demonstrate that the fitness plots at 20 and 24 hours are not qualitatively different, we have also shown the fitness plots at 24 hours for comparison. We have indicated in the figure caption that the additional four hours of growth merely allows the very low GAL-ON populations to reach saturation.

- To prove mutual invasibility, competition experiments between pure GAL-ON and GAL-OFF strategists were evaluated. Figure 2C shows that the fraction of GAL-ON and OFF strategists increases over time when initially rare. However, Figure S2B shows data from similar (or the same?) experiments. Here, a low initial GAL-ON frequency increases towards the end of the experiment, consistent with Figure 2C, but for high initial GAL-ON frequencies (i.e. rare GALOFF), the curves don't show a clear trend toward lower GAL-ON frequencies (i.e. more GALOFF) toward the end. According to Fig 2C, the downward shift in GAL-ON frequency is relatively small for this case (and also the difference in fitness according to Fig 2D), so maybe it is just a matter of scaling. The reader shouldn't be left alone to find out how these two plots can be reconciled.

We apologize for the confusion. The apparent contradiction in initial high GAL-ON frequencies between Fig 2C and Fig S2B is indeed a scaling issue combined with different initial starting fractions between the two experiments. Figure 2C does not have a pure GAL-ON population, whereas the topmost (darkest) line in Fig S2B is a population of pure (100%) GAL-ON, and thus cannot decrease in GAL-ON frequency over the course of the experiment. The next population (92% initial GAL-ON) does decrease in frequency. This is perhaps best visualized by Fig S2C, which shows change in frequency from initial. The magnitude of this change is basically consistent with that seen in figure 2C, though because of the different scaling and multiple points, it is less easy to discern from Figure S2B. Figure S2C is intended to make the decrease easier to discern.

To further address this concern, however, we have highlighted the fact that the darkest line in Figure S2B is 100% initial GAL-ON by adding the following italicized text to the Figure S2 legend: *"ten different initial fractions (from 0% to 100%) of GAL-ON over the course of a 20 hour incubation... With the exception of the 0% and 100% GAL-ON populations, the fraction of GAL-ON initially decreases relative to GAL-OFF (with around a 10% relative fitness cost..."*

- The caption of figure S2 says that colors indicate initial frequencies of the GAL-ON pure strategist. Most certainly, Figure S2D does not contain data for 0% and 100%, as GAL-ON

relative fitness cannot be calculated in this case. The GAL-ON percentages of the lightest curves in Figure S2D should be stated clearly so it is possible to read off the initial GAL-ON frequency that leads to a relative fitness of 1 (after 20h) from the graph. Also, in figure S2C, the 0% curve is probably covered by the black curve, so the same thing applies to this panel.

We thank the reviewers for pointing out the oversight in Figure S2D. Indeed, 0% and 100% are not shown. And, as pointed out by the reviewers, the 100% and 0% curves are overlaid on the x axis in Figure S2C. Accordingly, we have added the following statements to Figure S2D legend for clarification:

“In all plots, color gradients indicate the increasing initial frequencies of GAL-ON as described in panel A legend, except panel **D**, wherein the 100% and 0% initial GAL-ON populations are not shown because no relative fitness can be computed on a single strain. It should be noted also that in panel **C**, the 0% and 100% initial GAL-ON curves are shown, but are both overlaid on the x axis.”

- Starting on line 210 of the main text, GAL-OFF pure strategists are said to take long to transition to GAL-ON after glucose depletion, referencing figure S3 as evidence. However, in figure S3, only wild-type dynamics is shown. We couldn't find this evidence in any other figure.

We apologize for this error.. Figure S3 now demonstrates that GAL-OFF pure strategists do not transition to GAL-ON within the course of the 20 hour experiment.

*- When referring to Figure S1 on line 100 of the main text, it is not clear that the observed bimodal distribution of GAL is not just a transient state that is observed because some cells are a little bit faster switching to GAL-ON than others. Only together with figure S3 (and the plateau phase of GAL-ON in the intermediate galactose case in panel B), it becomes clear that this distribution is not just the snapshot of a transition process from GAL-OFF to ON. It would be optimal to either show two different time points for each plot in Figure S1 or to show that the bimodal activation even persists in an environment with *constant* intermediate levels mixed sugars.*

We agree that Fig S1 alone is indeed not sufficient to demonstrate that the GAL bimodal activation is stable over the initial phase of the experiment. In the main text, we have cited the extensive characterization of the stability of the bimodal GAL response in mixed glucose and galactose by Venturelli et al (2015) and Escalante et al (2015). We feel that these previous works are sufficient to establish that the bimodal GAL response is stable over time. We have added the following to Figure S1 caption: “Broadly speaking, the wild type W303 exhibits stable bimodal GAL expression over a wide range of roughly equal ratios of glucose and galactose concentration. (for in-depth characterization of the bimodal GAL response and its stability, see [23] and [25]. “

Minor issues:

- Line 193 and 196 of the main text refer to Figure 5, but only Figure 4 makes sense here.

Fixed.

- The line plots in Figure 2C are misleading because they hide the non-trivial time course of the GAL-ON fraction in the population that is shown in the more detailed Figure S2B. It looks like the fractions are steadily increasing/decreasing/constant, which is not actually the case. An alternative way of showing only beginning and end state could be to plot end state against initial frequency and draw an identity line for reference. At least, one should find a way to remove the lines that give a false impression of the time course.

This is an issue that we have wrestled with, as we believe that the simplest way to visualize negative frequency dependence is to see that low starting fractions increase and high starting fractions decrease. We very much like difference equations, but we have found that many readers have difficulty interpreting them. For this reason, we prefer to keep the two-timepoint graph. However, we have connected the points of measurement with dotted lines to make our point clearer, and to further prevent confusion we have added a sentence to the figure legend saying:

“Frequencies are shown at beginning and end for simplicity; temporal dynamics are explored in more depth in Appendix Figure S1”

- Line 121 says that the evolutionary stable mix is uninvasible by any phenotypic variant. At this point, it is not clear what "variants" are included here. Clearly, it cannot be any arbitrary phenotype. From the supplement, it seems the authors refer to either pure phenotypes or mixed strategies with different probabilities, but this should be explicitly mentioned.

We agree that this wording was unnecessarily vague. This sentence now reads: “An isogenic population that adopts the stable mix via phenotypic heterogeneity renders itself uninvasible by a mutant implementing either pure strategy or any probabilistic possible mix of the two”

- It is difficult to differentiate the darkest colors in Figure S2A.

We have expanded the color saturation range of Figure S2A to make it easier to distinguish the darker colors.

- The axis labels on the little plots in figure S1 are not readable. As they are probably arbitrary units, they can be removed and the figure caption could mention that the axis scaling of all the plots is identical.

We have removed the axis labels from Figure S1 and added to the figure caption that the axis scaling of the plots is identical.

- The figure caption for panel S3D is missing.

We have added the figure caption for panel S3D.

- The figure caption of figure S6 refers to figure 6. But there is no figure 6 in the main text.

We have changed it to refer to figure 5.

- The caption title (bold text) of figure S7 is identical to that of figure S6 and should be edited to reflect the actual facts.

Fixed .

- In the last line on page 3 of the Supplement, the word "fitness" is missing within the first brackets.

We have inserted the missing word into the equation

- On page 5 of the Supplement, the enumeration appears to be off by 1.

Fixed

- One page 9 of the Supplement, the second paragraph ends with an incomplete sentence.

We have finished the sentence: *.. so the optimal bet-hedging phenotype must feature phenotypes that have equal long-term fitness.*

- The last sentence on page 6 of the Supplement refers to Fig. 1. Shouldn't this be Fig 2? Also, the text says $V=2/3 C$, but the figure caption says $C=3V$.

The figure caption was in error. It now reads $C = 3/2 V$

Reviewer #2:

This paper documents how feeding on glucose versus galactose in yeast generates negative frequency dependent selection that leads to a stable equilibrium. Starting from yeast that can

feed on both glucose and galactose, the authors generated otherwise-isogenic strains that could grow only on glucose or only on galactose. Mixed at different frequencies, these showed all the hallmarks of negative frequency dependence and did not fit other explanations for the maintenance of two phenotypes. The paper is extremely thorough and includes theoretical models of how feeding on two foods can generate negative frequency dependence. It links, for perhaps the first time, frequency dependent selection with stochastic phenotype switching.

We appreciate reviewer #2's strong endorsement of our approach.

I have not major criticisms. I was a bit puzzled by the introduction, which takes as its starting point stochastic gene expression, which sometimes leads to coexistence of phenotypes. Frequency dependent selection comes in only later. I assume this is because it is the path that the authors own thinking followed. But I think most evolutionary biologists, when faced with the problem of stability of multiple phenotypes, would think of frequency dependent selection first and stochastic bet hedging somewhere down the list. Perhaps this is a difference in the microbial field, but even there, I think the authors underestimate the amount of work that has addressed frequency dependent selection and more of that work should be referenced (even if it is not related to stochastic expression).

This is not to take anything away from the substance of this paper. Though there are other papers on frequency dependence in microbes, this one is exceptionally rigorous and thorough. And it does make a link to stochastic gene expression that other papers do not make.

We agree that there were several approaches that we could have used in the introduction. The reason that we focus on phenotypic heterogeneity in clonal populations is twofold: 1) We believe that the community of researchers studying phenotypic heterogeneity in clonal populations has made a significant oversight in not including mixed strategies as a possible evolutionary driver, and 2) Negative frequency dependence has been observed in many other context between genotypes, so this is not as novel.

Section S1 of the supplement seems unnecessary to me as it mostly (entirely?) re-derives established results. It might be useful as a primer for those who are unfamiliar with bet-hedging or negative frequency dependence, but if it is kept for that reason, it needs to be referenced much more thoroughly.

We agree that Section S1 is in principle not necessary, but we thought that it would be useful for some readers to have this material easily accessible, to review some intuitive differences between stochastic bet-hedging and frequency-dependence. The editor has suggested that we move a condensed version of this section to a "box" in the main text. We believe that this will help make it more explicit that this material is not an original contribution on our part, and we have cited appropriate sources in the box.

Section S2 is more specific and more novel and it generates results important for understanding the kind of two-resource frequency dependence studied in the main paper.

Section S2 is indeed modeling that we have performed to elucidate what we believe are the essential dynamics present in our system.

Line 106. Should this be Figure S3 instead of S4?

This reference has been clarified.

Line 250. Figure 5 instead of Figure 6?

We have changed this reference to Figure 5.

Reviewer #3:

The goal of this manuscript is to test the hypothesis that negative frequency-dependent selection underlies phenotypic heterogeneity in microbial populations. This is examined in the yeast GAL

network that is expressed in a subpopulation of yeast cells when they grow in a mixture of glucose and galactose. This is a "mixed strategy" that is evolutionarily stable to "pure strategist" invaders that take up only glucose or only galactose. The manuscript starts by listing a set of criteria that distinguish negative frequency-dependent interactions from two other causes of phenotypic heterogeneity: bet-hedging and division of labor. Experiments are performed to check if the yeast GAL system fulfills some of these criteria. Finally, evolution experiments are performed to observe the conversion of pure strategists into mixed strategists, according to the expectation that once they arise, mixed strategists should invade pure strategists.

The manuscript poses an interesting, novel idea, and the experiments generate interesting data. It is not entirely clear, however, if the data fully support the claim that the yeast GAL network is a clear example of phenotypic heterogeneity arising from negative frequency-dependent selection. It is unclear if the other strategies can be excluded. The manuscript may be publishable if most of the following comments are addressed in a revised version.

We appreciate reviewer #3's clear summary of our claims and the generally positive evaluation. Below we address the specific concerns.

(1) The GAL gene expression distributions are recorded at a certain time during a 20-hour time course. However, during these 20 hours the cells run out of glucose, and even most of the galactose in the growth medium. Therefore, the glucose/galactose ratio will be entirely different at the end of the time course compared to the beginning. Consequently, the GAL gene expression distribution (the fractions of On and Off cells) will change as well such that in the end all cells may be GAL-On (as if they were pure strategists). These distributions should be measured during the 20-hour time course to determine how they change. How does this affect the model and the conclusions?

The reviewer is correct that the glucose and galactose concentrations in the media will change over the course of the 20hr time interval. We have found that there are essentially two phases of growth in our experiments. During the first phase GAL OFF cells grow more rapidly and the glucose becomes exhausted. At this point the GAL ON cells have an advantage, allowing them to grow in frequency. This basic dynamic can be observed in Figure S2. The frequency dependence we observe is largely a result of the sugar limitation described by the reviewer, since the more GAL-ON cells are present when glucose is depleted, the less expected growth per GAL-ON cell.

Additionally, the sugar depletion and consequent GAL activation of the "off" fraction in the wild-type W303 cells, can be seen in Appendix Figure S3. Appendix text 2 contains modeling and simulations that explore in detail how changes in the sugar concentration and activation of the GAL network effect the conclusions of the model.

(2) One underlying assumption is that cells do not switch from one subpopulation to the other. This is based on Fig. S4. However, Fig. S4 does not support this claim. In fact, cells sorted from the Off subpopulation seem to switch very quickly On (in a matter of hours). The "memory" of similar transitions has been measured before in yeast, and it can be up to days or even weeks (see PMID15889097 and PMID18362885). For example, in PMID18362885, switching is "slow" if it happens in ~100 hours or more, and even "fast" switching happens over more than 15 hours. Compared to these values, the switching rates in the GAL system actually seem super-fast. They should be measured exactly, in both directions (On to Off and Off to On) and incorporated in computational models to determine their effect. The rates should be determined based on the "Population dynamics model" in PMID18362885, to account for different growth rates of GALOn and GAL-Off cells.

The designation of "fast" vs "slow" depends upon the context. In our case, the glucose is exhausted after ~10 hours, which sets a natural timescale over which switching might be relevant. However, we recognize that this might be very fast relative to GAL switching in other contexts. In response to reviewer #3's observation to this effect, we have changed the wording of the text to read "a modest rate of stochastic switching between the states after the initial stochastic bimodal activation." (line 129).

In accordance with reviewer #3's suggestion, we have measured the switching rates more exactly (both OFF to ON and ON to OFF). We estimated KON and KOFF to be 0.08/hr. and 0.015/hr, respectively. This data is presented in the new Appendix Fig S5.

We have also incorporated stochastic switching into our computational modeling, and we explore the consequences of stochastic switching in Appendix Text S2 section IV. We find, perhaps unsurprisingly, that the strength of the negative frequency dependence decreases as the stochastic switching rate increases, since at higher stochastic switching rates, the initial ON and OFF distributions become more similar to each other for more of the duration of the competition.

(3) Mutual invasion of pure strategists: this probably depends on the initial % glucose and % galactose in the medium. That is, if there is ~ 1 or 2 orders of magnitude more galactose than glucose then GAL-Off purists will not reach high enough numbers at the end of the time course to be considered invading (simply because they do not have enough sugar to grow on). The reverse may also be true. This should be investigated more carefully and the manuscript should be revised accordingly.

In the limit that the glucose concentration goes to zero and the galactose concentration becomes large we indeed find that the GAL-ON mutant has a fitness advantage at all frequencies. This point is indicated in Figure 3C. To be more explicit that mutual invasibility is dependent on the ratio of sugar concentrations, at line 230 we have added "*Mutual invasibility is itself dependent on the ratio of sugar concentrations, since in the limit of a low glucose to galactose ratio, we observe that the GAL-ON phenotype is preferred at all frequencies.*"

(4) Fig. S2 has no error bars on any of the panels. Were these values measured in triplicate? If not, why not? If yes, are the error bars smaller than the symbols? This should be clearly stated. How exactly is Fig. 2C obtained from Fig. S2? This should be very clearly explained.

In Fig S2, initial frequencies (frequencies of GAL-ON and GAL-OFF before the 20 hr incubation in mixed sugars) were measured after the cultures been pre-mixed and grown for an overnight doxycycline induction/growth cycle. This induction cycle renders it difficult to start independent competitions at the same (post-induction) initial frequencies. Even when co-cultures were initially (pre-induction) mixed at identical frequencies, after the induction phase, the initial frequencies spread out significantly. As an illustration, the 60 post-induction initial frequencies shown in Figure 2 were actually 6 replicates mixed at 10 identical pre-induction frequencies, but those frequencies had spread out to a more or less continuous distribution (those displayed in Figure 2) during the induction growth phase. At one point we toyed with binning the initial frequencies in Figure 2 and displaying them with both vertical and horizontal error bars, but given the continuous distribution of frequencies, it was not obvious how they should be binned, and we concluded that plotting all the initial frequencies individually was more to the point anyway. Replicating the ten frequencies shown in Figure S2 would have the same result: there would be ten additional initial frequencies rather than replicates of the 10 shown. We felt that ten separate cultures at different starting frequencies were sufficient to demonstrate the trends we were measuring.

(5) The models presented in the S1 Text are very simplistic game-theoretic models. It would be helpful to try other models, especially for bet-hedging. For example, the differential equation model in PMID18362885 (or its stochastic implementation) would be very useful to try. The conclusions should be robust to the type of model used. Moreover, more experimental details should be incorporated into models. At least some of the models should be more than simplistic illustrations of ideas: they should be rooted in and should be constrained by experimental observations.

The models in S1 are intentionally simplistic, and were meant to explain the most basic versions of these models. To highlight the role that this section was meant to play, the editor has suggested that we move a condensed version of this section to a "box" in the main text. In contrast to the simplistic and intuitive models of S1, supplementary text S2 contains in-depth modeling of microbial cultures growing in two-sugar finite resource environments, and incorporates more experimental details.

Following referee #3's suggestion, we have explored the model described in *PMID18362885*, which we present in detail below. In particular, the bet-hedging model in this paper has no possible mechanism for frequency dependence. For example, the ON type has a fixed growth rate in each environment that is independent of the rest of the population, meaning that the growth rate averaged over the environmental fluctuations will also be independent of the rest of the population. Nevertheless, we have performed simulations of this model to confirm that there is no frequency dependence. In particular, we consider the situation in which there are two competing genotypes, each with different switching rates in the ON and OFF directions. Although the fluctuating environments cause the population size of the two types to fluctuate, the number of divisions over the first 60 hr is independent of the starting frequencies. In left figure N population in solid and M population in dashed, whereas in the right figure N population in dark grey and M population in light grey.

Switching rates for N:

$$k_{on} = 0.1;$$

$$k_{off} = 0.1;$$

Switching rates for M:

$$r_{on} = 0.01;$$

$$r_{off} = 0.01;$$

Environment 1 (environments alternate every 5 hr):

$$dN_{off}/dt = N_{off} - k_{on} * N_{off} + k_{off} * N_{on}$$

$$dN_{on}/dt = -N_{on} - k_{off} * N_{on} + k_{on} * N_{off}$$

$$dM_{off}/dt = M_{off} - r_{on} * M_{off} + r_{off} * M_{on}$$

$$dM_{on}/dt = -M_{on} - r_{off} * M_{on} + r_{on} * M_{off}$$

Environment 2:

$$dN_{off}/dt = -1.5 * N_{off} - k_{on} * N_{off} + k_{off} * N_{on}$$

$$dN_{on}/dt = 0.5 * N_{on} - k_{off} * N_{on} + k_{on} * N_{off}$$

$$dM_{off}/dt = -1.5 * M_{off} - r_{on} * M_{off} + r_{off} * M_{on}$$

$$dM_{on}/dt = 0.5 * M_{on} - r_{off} * M_{on} + r_{on} * M_{off}$$

(6) There was no sequencing performed to confirm whether or not the phenotypic mix of GALON and GAL-OFF observed in the evolution experiments (e.g., the mixed "clonal" populations presented in Fig. 5) is the result of a genetic or epigenetic/nongenetic phenomenon. If the changes are in fact nongenetic, then it is debatable whether or not evolution actually occurred. If there are mutations occurring, then which type, where in the GAL network/genome, and if multiple mutations occurred in what order did they occur and is this important? This being the case, further experimentation/sequencing should be performed to elucidate the underlying mechanisms of phenotypic variability in the GAL network.

We agree that it would be very interesting to determine the molecular origin of the change in phenotype that we observed in our evolution experiments. However, given the large number of different evolved populations and the large number of possible locations at which the mutations (assuming that they exist) might have taken place, we believe that sequencing of the evolved

populations is beyond the scope of this project. Indeed, even if the changes in the population were the result of an epigenetic/non-genetic phenomenon, it would nevertheless constitute a heritable change in the population that leads to stable coexistence of the two phenotypes. This is the primary point that we would like to argue, and is independent of the detailed molecular mechanism.

(7) One interesting experiment to run is to compete a mixture of the "pure" strategists with the mixed strategist. That is, a 3-strain competition. Would the mixed strategist invade the mixture of the pure strategists or vice versa?

We agree that this would be an interesting experiment. Unfortunately, we are already using the three fluorescent “channels” on our flow cytometer that allow us to distinguish different cell types, so we are not able to perform 3-strain competitions. We expect that the wildtype “mixed” strategist would invade most mixtures of the two pure strategists (although the simplest model of mixed equilibria predicts that the mixed strategist should not be able to invade the two pure strategists if they are already at equilibrium).

Minor comments:

(1) It was mentioned that the study is "in the context of an exponentially growing microbial population", however cell populations were allowed to reach saturation? Why were the cells not maintained in exponential growth phase by using more frequent dilutions of larger cell culture volumes? At saturation, starvation and other responses in budding yeast may be activated which makes for a less controlled experiment.

We apologize that we were not clearer in our wording of the experimental setup. The statement about the exponential growth context was brought up as a reason for which it may not be intuitively obvious to expect a culture of growing microbes in media to conform to the simplistic intuition about mutual invasibility given by frequency-dependence models like foraging game described in Figure 1 A&B and the hawk-dove model described in Box 1, neither of which feature generations of population growth over the timeframe of the game. Our point was that, in considering a microbial foraging game—in which microbes grow exponentially on a timeframe less than that of the resource depletion—some more explicit modeling might be needed to gain intuition about frequency dependent selection. This modeling is presented in Figure 1 C&D and supplementary text S2.

To clarify this point, we have modified the sentence at line 158 of the main text to read:

“However, this simple foraging game becomes slightly more complicated in the context of a microbial population, which may undergo generations of exponential growth over the course of the resource depletion. To investigate the fitness dynamics that might result from this more complicated foraging game, we modeled a microbial population growing in the presence of two resources...”

Regarding allowing microbes to reach saturation, we were specifically interested in the case wherein a finite supply of two different sugars causes frequency dependent selection and mutual invasibility between two phenotypes which specialized in consuming the one or the other, so running out of resources is central to the experimental setup. However, in Appendix text S2 we do model a simplifying assumption wherein exponential growth is maintained till the end without a saturating growth phase, to satisfy ourselves that frequency dependence can theoretically be observed in either case.

(2) Apparently, the peak positions shift in Fig. S1. How would this affect the models and the conclusions?

The peak positions do shift in Fig S1, with higher galactose (and lower glucose) producing more GAL proteins per cell. Intuitively, this could affect the models and conclusions in two ways: higher GAL activation could raise the metabolic fitness “cost” of the GAL network, making GAL-ON a relatively less desirable phenotype than it otherwise would be. Higher GAL activation could also come with a faster consumption of galactose, and concomitant fitness

boost. Within Appendix text S2, variable metabolic cost and variable consumption rates of galactose are explored on Figure 3 and Figure 6, respectively. Both of these parameters affect the conclusions of the model as shown, but in general we find that our conclusions are fairly robust to these effects.

(3) It was not clear immediately why a higher growth penalty for phenotype B results in a higher phenotype B fraction in the growth optimal mix? Perhaps the authors can provide a more intuitive explanation for the reader.

We agree that this is a somewhat counterintuitive result and we should have provided more intuition. On line 173 we have now written the following: “Indeed, with increasing growth cost the two solution concepts diverge (Fig 1D). As the cost incurred by phenotype B increases, the growth-optimal fraction of this phenotype increases, as the population needs to devote more individuals to this phenotype in order to consume the two resources at the same time (which maximizes the fitness of the population). However, at this growth-optimal fraction the individuals with phenotype B have lower fitness, making this fraction evolutionarily unstable.”

(4) Top of Page 5: I think the text tries to refer to Fig. 4.

We have clarified this reference.

(5) The manuscript would benefit from the elimination of some jargon and game-theory terminology.

We apologize for the jargon and game-theory terminology. Some of this is perhaps required for precision in explaining why mixed game strategies are different from other probabilistic strategies. However, we have gone through the text with this comment in mind. For example, we have replaced several instances of game theoretic specific words like “payoff” with more traditional evolutionary fitness terms (see line 211, figure 1 caption, and figure 3 caption). We have also added explanations where confusion may arise. For example, at line 249, to clarify the meaning of neutral uninvasibility .

“In other words, in the limit of a population consisting entirely of mixed strategists, any single invading cell adopting any strategy (pure or mixed) will have the same fitness as the mixed strategist, and the mixed strategist might then be termed *neutrally uninvasible* at high frequencies. “

2nd Editorial Decision

10 December 2015

Thank you again for submitting your revised work to Molecular Systems Biology. We have now heard back from the two referees who agreed to evaluate your manuscript. As you will see from the reports below, while reviewer #1 is satisfied with the modifications made, reviewer #3 still raises significant concerns on your work which, I am afraid to say, preclude its publication in Molecular Systems Biology.

In particular, reviewer #3 raises substantial concerns on the experimental system chosen and is not convinced that the presented analyses conclusively support that negative frequency-dependent interactions underlie phenotypic heterogeneity. Overall, this referee is not persuaded that the issues raised previously have been satisfactorily addressed in this revision and s/he points out that the study is not suitable for publication in Molecular Systems Biology.

Taken together, considering the substantial concerns raised by reviewer #3, and in combination with the fact that our editorial policy allows in principle a single round of major revision, I see no other choice than to return the manuscript with the message that we cannot offer to publish it.

REFEREE REPORTS

Reviewer #1:

The authors made a good effort to address our concerns and concerns of other reviewers. We believe that the paper is now acceptable for publication in MSB. There is one typo which needs to be corrected however. On line 220 the reference evidently should be to the figure S4, not S3.

Reviewer #3:

I have read the manuscript multiple times. While some minor editing has slightly improved the clarity of the manuscript, it still remains confusing. Some models were added to the Appendix. The work invested to address the comments is not extensive.

Overall, I am not convinced that this work identifies a biological system in which negative frequency-dependent selection underlies phenotypic heterogeneity. I am sure that the effect is possible - but a different system may need to be found that convincingly supports the claim that negative frequency-dependent selection underlies phenotypic heterogeneity.

The main problem is the choice of system that is in a "gray zone" between typical hawk-dove type models and models of bet hedging. On the one hand, frequency-dependent selection and social games (like hawk-dove) are classically investigated in constant environments. On the other hand, bet-hedging is defined in environments that fluctuate repeatedly, many times. Yeast cells growing in a mixture of glucose and galactose do not fit in either of these categories. The environment is clearly not constant - since the fitness and distribution of phenotypes change drastically as glucose is depleted. Therefore, this experimental system does not fit the classical Hawk-Dove model. On the other hand, the environment does not fluctuate repeatedly either. Therefore, the experimental system is not a typical example for testing bet-hedging either. It is in a gray zone between the two, making the claim that negative frequency-dependent selection underlies phenotypic heterogeneity unconvincing.

Three unique predictions are listed that distinguish negative frequency-dependent selection from bet-hedging as the source of phenotypic heterogeneity. Unfortunately I am not convinced that these features apply to this experimental system sufficiently to decide between bet hedging and frequency-dependent selection.

The first criterion specific to frequency-dependence-driven mixed ESS is mutual invisibility of pure phenotypes. "With frequency-dependence-driven mixed ESS, pure phenotypes are mutually invisable, while in bet-hedging they are not." While this may be true after many environmental shifts, I am not convinced that pure phenotypes are not mutually invisable in bet-hedging when the environment shifts only once (as it happens here when glucose is depleted). I can imagine a mixture of slow- and fast-growing cells (S and F) that start growing in a good environment (G) that later turns bad (B). Imagine that the cells grow at different rates in environments G and B, and they also switch randomly between phenotypes S and F. Then one could imagine that at a specific time T^* after the environment shifts to B, the frequencies of S and F phenotypes are reversed compared to their initial fractions. If the system starts with mostly S cells then at T^* the frequency of S cells decreases. If the system starts with mostly F cells then the opposite occurs at time T^* . Unless it is proven that this is impossible for any parameter combination and for any choice of T^* , bet-hedging cannot be excluded as the cause of phenotypic heterogeneity in this experimental system. Whether pure phenotypes invade each other depends not only on the glucose/galactose ratio, but also on the choice of time T^* in the current system. Therefore, I am not convinced that the first criterion can distinguish between the two scenarios in the current system.

The second criterion is equal fitness at the stable point. Once again, the definition of fitness depends on the time of measurement. If a different time is chosen, the fitnesses may not be equal. Moreover, even at the chosen time of measurement, in the current system one pure strategist does not display

the neutral uninvasibility feature. Therefore, the second criterion does not really apply to the current system.

The third criterion, suboptimal yield at the stable point should be specific for ESS versus optimal phenotype distribution in bet-hedging - but only if the environment shifts many times. General statements of optimality cannot be made after a single environmental shift. Moreover, in my opinion bet hedging cannot be restricted to the optimal phenotype distribution. A bet-hedging cell population is one where cells assume at least two different phenotypes such that their overall fitness is greater than that of either pure phenotype after many environmental shifts. Of course, evolution may result in bet-hedging populations eventually approaching the growth-optimal phenotype distribution. But growth optimality should not be part of the definition of bet hedging. It would make the definition extremely restrictive and perhaps unrealistic. In fact, the non-optimal evolutionarily stable mix could safely classify as bet-hedging if it grows faster than either pure phenotype after many shifts from glucose+galactose to galactose (repeated resuspension in glucose+galactose). Therefore, the third criterion does not seem valid.

Recently, Venturelli et al. (ref. [38]) have examined phenotypic heterogeneity in similar conditions. They have measured fitness, and have found bistability in a biochemical model that underlies the phenotype heterogeneity. Taken at face value, the novelty of the current manuscript may consist of experiments testing invisibility of strategies and experimental evolution. However, without convincing arguments for ESS over bet hedging, I am not sure if the data alone justify publication in MSB.

Specific comments:

(1) Page 5, lines 108-118: specify the main features of the real system that make it different from this and other models. This was a request by two reviewers. For example, galactose is not consumed by the GAL-ON phenotype while glucose is present. While most of this is in the Appendix, it should really be in the main text. Same for page 6, lines 129-134.

(2) S1 text: Table 1 is not a good illustration of bet-hedging. A completely homogeneous population could have these individual, combined and mean yields. There is nothing about bet-hedging in this model.

(3) S1 text, page 3: The optimization of geometric mean fitness is an old concept that needs some clarification when applied to exponentially growing populations. Specifically, if fitness is defined as the exponential growth rate then the arithmetic (not geometric) mean of growth rates is optimized for populations growing exponentially in fluctuating environments.

(4) S1 text, Pages 5-7: this is the classical Hawk-Dove model in a constant environment. Notation should indicate that the expected payoff $E(I)$ depends explicitly on the frequency of each phenotype. Moreover, it could also depend on time when resources are consumed.

(5) Only simple classical models are presented in S1. S1 could be made clearer if an intermediate model is also presented that turns into bet-hedging or ESS as some parameters are changed.

(6) S2 text, pages 12-15: While this model apparently shows negative frequency dependence, it is not truly relevant to the experimental system because the phenotypes M and N do not interact at all. Cells of type M and N could as well be grown in separate flasks. This means that negative frequency dependence does not require any direct or indirect interactions between phenotypes. How would it drive phenotypic heterogeneity then? Each population could just be growing separately and remain homogeneous. Cells may run out of galactose before glucose in this model. These limitations should be clearly stated. Moreover, negative frequency dependence will depend on the parameters (growth rates and initial resource levels). This should be stated and perhaps studied.

(7) S2 text, pages 20-23: Equations are missing for this model.

(8) S2 text, pages 26-27: This is the most relevant model. However, it is presented too briefly. The switching rate (at least k_{ON}) should be sugar-dependent to make this model even more similar to the real system.

Second submission

25 April 2016

Reviewer #1

The authors made a good effort to address our concerns and concerns of other reviewers. We believe that the paper is now acceptable for publication in MSB. There is one typo which needs to be corrected, however, On line 220 the reference evidently should be to the figure S4, not S3.

We are glad that referee #1 feels that we have properly addressed both his/her concerns as well as the concerns of the other referees.

The typo on line 220 has now been fixed. Thank you for catching it.

(Continued on next page.)

Reviewer #3

I have read the manuscript multiple times. While some minor editing has slightly improved the clarity of the manuscript, it still remains confusing. Some models were added to the Appendix. The work invested to address the comments is not extensive.

Overall, I am not convinced that this work identifies a biological system in which negative frequency-dependent selection underlies phenotypic heterogeneity. I am sure that the effect is possible - but a different system may need to be found that convincingly supports the claim that negative frequency-dependent selection underlies phenotypic heterogeneity.

The main problem is the choice of system that is in a "gray zone" between typical hawk-dove type models and models of bet hedging. On the one hand, frequency-dependent selection and social games (like hawk-dove) are classically investigated in constant environments. On the other hand, bet-hedging is defined in environments that fluctuate repeatedly, many times. Yeast cells growing in a mixture of glucose and galactose do not fit in either of these categories. The environment is clearly not constant - since the fitness and distribution of phenotypes change drastically as glucose is depleted. Therefore, this experimental system does not fit the classical Hawk-Dove model. On the other hand, the environment does not fluctuate repeatedly either. Therefore, the experimental system is not a typical example for testing bet-hedging either. It is in a gray zone between the two, making the claim that negative frequency-dependent selection underlies phenotypic heterogeneity unconvincing.

We would like to highlight that the simple foraging game described in Figure 1 is definitely not a bet-hedging situation, despite the fact that the resources are depleted by the players. Therefore, in our mind there are situations that do contain a changing environment that can only be described as negative frequency dependent interactions leading to mixed strategies (or coexistence of pure strategists). Indeed, outside the specific context of clonal phenotypic heterogeneity, negative frequency dependent selection (as a driver of stable genetic polymorphism, for example) is not uncommonly a result of the fact that an overabundance of some specialist phenotype/genotype/species depletes some niche resource faster, leading to invasion of minority phenotypes/genotypes/species, and ultimately stable coexistence. In this paper, we merely propose that similar evolutionary forces may promote the same phenotypic coexistence but in clonal microbial populations.

Nevertheless, to address the concern of Reviewer #3 we have performed new experiments in pseudo-continuous culture, in which we are constantly replenishing resources (as compared to the primary experiments that are done in batch culture, where the sugars concentrations change significantly over a 20 hour period of competition). Specifically, we took saturated co-cultures of the GAL-ON and GAL-OFF strategists and competed them in "continuous" culture in which we diluted the co-cultures every 3 hours into fresh media, thus mimicking a chemostat that operators at constant dilution rate. The sugar concentrations and cell densities therefore reached an equilibrium, as the sugars were being constantly replenished. In these experiments, we observe strong negative frequency dependent

interactions, with the relative fitness crossing one (indicating stable coexistence). Importantly, this negative frequency dependence was not observed in pure glucose conditions. These experiments are now included as supplementary figure S11.

Accordingly, at line 167 we have added the following paragraph:

“We therefore observe negative frequency dependence during batch co-culturing of the GAL-ON and GAL-OFF strains due to exhaustion of resources, similar to the negative frequency dependence that is expected in the simple foraging game described in Figure 1. However, given that this mechanism of negative frequency dependence involves a changing environment, it superficially resembles the changing environments that are present in bet-hedging models. To demonstrate that the negative frequency that we observe does not rely on a changing environment, we therefore co-cultured the GAL-ON and GAL-OFF in pseudo-continuous culture. In particular, we diluted the co-culture by a factor of 2X every three hours, thus mimicking a chemostat operating at constant dilution factor. In these experiments we found strong negative frequency dependent selection between the GAL-ON and GAL-OFF strains when cocultured in a glucose+galactose environment, but no frequency dependence in a pure glucose environment (Fig S11). Negative frequency dependence between the GAL-ON and GAL-OFF strains can therefore be observed in either batch culture, in which resources are depleted, or in continuous culture, in which nutrients are continually being supplied.”

Supplementary Figure S11: Pure strategists show negative frequency dependence under quasi-continuous culture conditions in mixed sugar environment but not pure sugar environment. Pure strategists were mixed in a range of initial fractions in .01% glucose for 24 hours. They were then grown to saturation for 17 hours in either pure glucose (blue) or a mixed sugar environment (red). At $t=0, 3, 6,$ and 9 hours, the cells were diluted 2X to replenish with fresh sugars (blue: 0.03% glucose, red: 0.03% glucose + 0.02% galactose). At $t = 0, 3, 6, 9,$ and 12 hours, the fractions of the two strains were measured using flow cytometry. Relative fitness of the two strains is computed using the relative fraction of the two strains at $t=0$ and $t=12$ hours. The relative fitness of the galactose strategist is plotted as a function of its initial fraction. Error bars were determined using bootstrap.

Three unique predictions are listed that distinguish negative frequency-dependent selection from bet-hedging as the source of phenotypic heterogeneity. Unfortunately I am not convinced that these features apply to this experimental system sufficiently to decide between bet hedging and frequency-dependent selection.

The first criterion specific to frequency-dependence-driven mixed ESS is mutual invisibility of pure phenotypes. "With frequency-dependence-driven mixed ESS, pure phenotypes are mutually invisable, while in bet-hedging they are not." While this may be true after many environmental shifts, I am not convinced that pure phenotypes are not mutually invisable in bet-hedging when the environment shifts only once (as it happens here when glucose is depleted). I can imagine a mixture of slow- and fast-growing cells (S and F) that start growing in a good environment (G) that later turns bad (B). Imagine that the cells grow at different rates in environments G and B , and they also switch randomly between phenotypes S and F . Then one could imagine that at a specific time T^ after the environment shifts to B , the frequencies of S and F phenotypes are reversed compared to their initial fractions. If the system starts with mostly S cells then at T^* the frequency of S cells decreases. If the system starts with mostly F cells then the opposite occurs at time T^* . Unless is it proven that this is impossible for any parameter combination and for any choice of T^* , bet-hedging cannot be excluded as the cause of phenotypic heterogeneity in this experimental system. Whether pure phenotypes invade each other depends not only on the glucose/galactose ratio, but also on the choice of time T^* in the current system. Therefore, I am not convinced that the first criterion can distinguish between the two scenarios in the current system.*

We believe that the reviewer is confused by our use of the term "pure phenotype." We have used the phrase "pure phenotype" to indicate a strategy of adopting a single phenotype, as opposed to a strategy of adopting multiple phenotypes. This would preclude strategies like the one the reviewer outlines, which implement random switching between phenotypes. "Strategies" and "phenotypes" are often conflated in the literature (indeed, people speak of "bet hedging phenotypes"), but since the distinction is relevant here, a more precise term for "pure phenotype" in

our paper would be the term “pure strategy.” Our claim is that in the frequency-dependent scenarios that drive a mixed ESS, *pure strategies* are mutually invisable, whereas in fluctuating environments that drive bet-hedging, *pure strategies* are not mutually invisable.

We claim that there is no mutual invisibility of pure strategies in a bet-hedging model such as that presented by Acar et al in Nature Genetics (2008). In the Reviewer’s example above, “phenotypes” correspond to S or F. A *strategy* corresponds to a pair of switching rates between these phenotypes. In the case of pure strategists (no switching), the strategies and the phenotypes are the same thing. However, in the example that the Reviewer describes they are different.

We note that in terms of phenotypes there is apparent “invasibility” as a result of switching between the phenotypes, even in a constant environment and absent any frequency dependent interactions at all. A population starting all S will become a mixture of S/F, and a population all F will become a mixture of S/F. This apparent “invasibility” is only present in terms of phenotypes. *Strategies* in the Acar model are non-interacting (the growth rates are not a function of the population composition so there is no mechanism for the strategies to affect each other).

In reviewing our text, we can see that some imprecise wording we have used in our paper has contributed somewhat to this confusion (we had previously received the request to not use game theory “jargon” such as the phrase “strategy”, but in some cases this is necessary to avoid confusion). We have accordingly altered the paragraph explaining the simple foraging game to read as follows, changes in italics:

“Frequency dependent interactions may be especially relevant to multi-resource environments because of the simple foraging game that exists when resources are limited. For instance, consider an isogenic population that is confronted with a phenotypic decision to specialize in consuming one or the other of two limited food sources, A and B (Fig 1A). *We will define two available phenotypes as “specialize in A” or “specialize in B”. We will define a strategy as some heritable set of instructions about which phenotype to adopt.* The more individuals that adopt the *strategy “always specialize in A,” (a pure strategy)* the more quickly A will be consumed, reducing payout to individuals who choose that strategy. Hence, if all individuals choose “*always specialize in A,*” a mutant *pure strategist* that *always specializes in B* will have an evolutionary advantage, and vice versa. The *two pure strategies* may thus be *considered mutually invisable*, with an equilibrium consisting of a stable mix of the two (Fig 1B). *In theory, a clonal population may achieve the same stable mix of phenotypes by adopting a stochastic mixed strategy, wherein each individual adopts one or the other phenotype with some probability, either by making a single random choice or stochastically switching between the two phenotypes. Either way, an isogenic population that adopts the stable mixed strategy via phenotypic heterogeneity renders itself uninvasible by a mutant implementing either pure strategy or any other possible stochastic mix between the two.”*

In addition, we have replaced descriptions of phenotypes with descriptions of strategies in the following lines: 135, 176, 177, 181, 190, 233, 262, 265, 324, 365, 367, 369

Hopefully these changes help clear up the confusion.

The second criterion is equal fitness at the stable point. Once again, the definition of fitness depends on the time of measurement. If a different time is chosen, the fitnesses may not be equal. Moreover, even at the chosen time of measurement, in the current system one pure strategist does not display the neutral uninvasibility feature. Therefore, the second criterion does not really apply to the current system.

Fitnesses (and relative fitnesses) will always be a function of the environment (time included). If we let the experiment run for 24 hours instead of 20 hours then we would see a (slightly) different equilibrium between the pure strategists, but at equilibrium the fitness of two strains would still be the same (if they were not the same then the fraction would change until they had the same fitness). It is true that one of the pure strategists does not display the simple neutral invasibility with the wildtype that is expected from the most simple model. However, this was the only failure of many predictions from this simple model, so in our opinion the bulk of the evidence is consistent in providing support for the interpretation that it is negative frequency dependent interactions that are driving phenotypic heterogeneity in this system (with the caveat that it is of course impossible to prove the historical evolutionary driver for this or any other phenotype in wild populations... however we believe that our experimental evolution of mixed strategies in the laboratory provide compelling evidence for our interpretation).

The third criterion, suboptimal yield at the stable point should be specific for ESS versus optimal phenotype distribution in bet-hedging - but only if the environment shifts many times. General statements of optimality cannot be made after a single environmental shift. Moreover, in my opinion bet hedging cannot be restricted to the optimal phenotype distribution. A bet-hedging cell population is one where cells assume at least two different phenotypes such that their overall fitness is greater than that of either pure phenotype after many environmental shifts. Of course, evolution may result in bet-hedging populations eventually approaching the growth-optimal phenotype distribution. But growth optimality should not be part of the definition of bet hedging. It would make the definition extremely restrictive and perhaps unrealistic. In fact, the non-optimal evolutionarily stable mix could safely classify as bet-hedging if it grows faster than either pure phenotype after many shifts from glucose+galactose to galactose (repeated resuspension in glucose+galactose). Therefore, the third criterion does not seem valid.

We agree that a non-optimal bet hedging strategy is still a bet-hedging strategy. Our purpose is to remind the reader that—absent some other evolutionary driver such as frequency dependent selection—a bet-hedging strategy in principle evolves

toward a mix of phenotypes that maximizes population fitness in the environment in which it is evolving. As the reviewer points out, a clonal population may indeed implement a non-growth-optimal mix of phenotypes (and we may, as the reviewer suggests, still consider it to be fulfilling a bet-hedging strategy if the overall population fitness would be higher than either phenotype alone after many environmental shifts). But such a non-optimal “bet-hedging” strategy would not exist as an evolutionary equilibrium if the only evolutionary driver in play were the shifting environment. Such a strategy would be susceptible to invasion by a mutant that implements a phenotypic mix that provides a more optimal long-term population growth. (The same is not true in the case of a frequency dependent game like the hawk-dove game, in which the higher-fitness Always-Dove population of pure strategists fails to invade the less-fit population composed of the stable mix of hawks and doves.)

If the evolutionarily stable mix of phenotypes in a clonal population is *not* growth optimal, then, it indicates that some other factor besides just shifting environments is in play (such as negative frequency dependent interactions). Thus, though it may technically fulfill the role of a bet hedging strategy, it should not be considered *exclusively* a bet-hedging strategy, because if it were *just* the alternating environmental states driving the evolution of phenotypic heterogeneity, then the stable strategy would maximize population growth over time. Therefore, the observation of non-optimal evolutionary equilibria is an indication that at least one other factor is driving the heterogeneity besides shifting environments.

In any case, our paper is not primarily concerned with ruling out that heterogeneity in the GAL system in yeast can ever fit the definition of a bet-hedging strategy (indeed, given the definition of bet-hedging provided by reviewer 3, ruling out bet hedging in even the simplest hawk-dove-like foraging game presented in Figure 1 would be very difficult.)

Instead, our focus is in demonstrating that the system fits the expectations of an overlooked alternative explanation. Unlike with bet-hedging, in a hawk-dove-like foraging game, evolution does not necessarily drive the system towards a growth-optimal stable point. We observe that the equilibrium between the GAL-ON and GAL-OFF strategists in our system does not maximize fitness. Granted, this is not proof that the phenotypic heterogeneity in the wild type evolved in response to this frequency dependence (particularly since we don’t know the environment that our yeast evolved in); however, in our opinion, this result—along with our demonstration of negative frequency dependent selection between the two pure strategies—does provide compelling evidence in favor of mixed strategies as a relevant evolutionary explanation for such phenotypic heterogeneity.

To address the potential complexities that the reviewer has raised, we performed competition experiments in continuous culture, rather than the single finite-sugar consumption environment of figure 2) to demonstrate that the GAL-ON and GAL-

OFF strategies still display negative frequency dependence.

Recently, Venturelli et al. (ref. [38]) have examined phenotypic heterogeneity in similar conditions. They have measured fitness, and have found bistability in a biochemical model that underlies the phenotype heterogeneity. Taken at face value, the novelty of the current manuscript may consist of experiments testing invisibility of strategies and experimental evolution. However, without convincing arguments for ESS over bet hedging, I am not sure if the data alone justify publication in MSB.

We are of course familiar with the work of Venturelli et al. The biochemical network studied by these authors (and others, including Acar in his Nature paper in 2005) constitutes a mechanism of implementing a stochastic strategy. We believe that our manuscript contributes to this literature by exploring possible evolutionary drivers for a population to evolve a stochastic strategy, and in particular highlighting a largely-overlooked possibility that such stochastic strategies may arise in the context of negative frequency dependent selection between pure strategists.

Specific comments:

(1) Page 5, lines 108-118: specify the main features of the real system that make it different from this and other models. This was a request by two reviewers. For example, galactose is not consumed by the GAL-ON phenotype while glucose is present. While most of this is in the Appendix, it should really be in the main text. Same for page 6, lines 129-134.

We agree that there are multiple features in our experimental system that deviate from the most simple models. To address this we have added the following to line 190:

“A more in-depth investigation of the dynamics between the pure strategists indicates that while our system differs in several respects from the simulated models, the negative frequency dependence we observe is indeed related to the depletion of resources in the media. Unlike in the simulated models where each specialist strategy only consumes its niche resource, we find that in the presence of glucose and galactose both pure strategist strains consume primarily glucose while it is present (though GAL-ON cells consume some galactose during that time as well (Fig S2E). This observation is consistent with previous observations of the GAL network [36,43].”

(2) S1 text: Table 1 is not a good illustration of bet-hedging. A completely homogeneous population could have these individual, combined and mean yields. There is nothing about bet-hedging in this model.

We were attempting to illustrate the concept of geometric mean in the context of “conservative” bet-hedging (also known as “playing it safe”) in contrast to “diversified” bet-hedging (also known as “spreading the risk”). However, we agree

that this may lead to unnecessary confusion, so we have removed this table.

(3) S1 text, page 3: The optimization of geometric mean fitness is an old concept that needs some clarification when applied to exponentially growing populations. Specifically, if fitness is defined as the exponential growth rate then the arithmetic (not geometric) mean of growth rates is optimized for populations growing exponentially in fluctuating environments.

We agree that this is confusing. In the supplement we have added the following clarification:

“It is important to note that this maximization of geometric mean is defined in the present context where fitness is measured in discrete time as reproductive yield. However, in an exponentially growing population, where fitness is commonly defined as the exponential growth rate, then the arithmetic (not geometric) mean is optimized for populations growing in fluctuating environments.”

(4) S1 text, Pages 5-7: this is the classical Hawk-Dove model in a constant environment. Notation should indicate that the expected payoff $E(I)$ depends explicitly on the frequency of each phenotype. Moreover, it could also depend on time when resources are consumed.

This is indeed the classical hawk-dove game, in which there is no notion of time before which the resources are consumed. We had previously written:

$$E(I) = f * E(I,H) + (1-f) * E(I,D)$$

and we now add an explicit function of f :

$$E(I,f) = f * E(I,H) + (1-f) * E(I,D)$$

We hope that this helps the reader remember that the payout of a strategy I depends upon the frequency of the strategies in the population.

(5) Only simple classical models are presented in S1. S1 could be made clearer if an intermediate model is also presented that turns into bet-hedging or ESS as some parameters are changed.

We agree that this could be desirable, but we have received many conflicting requests regarding the supplement and we would prefer not to make this change as we believe that it would make other readers less happy. Our goal in S1 was just to provide interested readers the classical background in one place.

(6) S2 text, pages 12-15: While this model apparently shows negative frequency dependence, it is not truly relevant to the experimental system because the phenotypes M and N do not interact at all. Cells of type M and N could as well be grown in separate flasks. This means that negative frequency dependence does not require any direct or indirect interactions between phenotypes. How would it drive phenotypic

heterogeneity then? Each population could just be growing separately and remain homogeneous. Cells may run out of galactose before glucose in this model. These limitations should be clearly stated. Moreover, negative frequency dependence will depend on the parameters (growth rates and initial resource levels). This should be stated and perhaps studied.

This simple model is the microbial equivalent of the simple foraging game depicted in Figure 1 of the main text. In both cases the phenotypes are not interacting, and the food resource is split among the population that “chooses” to specialize in that food source. Although it may appear that the cells do not interact at all, in fact there is an interaction if the cells are choosing which phenotype to be. Just as in the simple foraging game, each individual benefits from having a larger fraction of the population choosing the opposite phenotype. This is what leads to negative frequency dependence in this simple model. In particular, a population growing on just one of the sugar sources can be invaded by a mutant genotype that specializes on the other sugar source (or that stochastically mixes between the two phenotypes).

(7) S2 text, pages 20-23: Equations are missing for this model.

Added.

(8) S2 text, pages 26-27: This is the most relevant model. However, it is presented too briefly. The switching rate (at least k_{ON}) should be sugar-dependent to make this model even more similar to the real system.

We apologize that our description of this model was not clear. In the simulations in which there was switching, k_{ON} was indeed sugar dependent. As we wrote:

“ $k_{on} = 0$ for the no switching condition

$$k_{on} = (0.5 \text{ hr}^{-1}) \frac{[gal]}{[gal] + C[gluc]}$$

$C = (125,50)$ for the intermediate and rapid switching conditions”

In the limit of large galactose concentrations the OFF cells rapidly turn on, whereas for low galactose concentrations the cells have negligible switching into the ON state.

To clarify this, we have explicitly mentioned the sugar-dependent switching in the opening paragraph of the section as well as in the figure legend. The starting paragraph now reads (note italics):

“Unlike the GAL-OFF and GAL-ON pure strategies, the wildtype yeast cells are able to stochastically switch between the GAL ON and GAL OFF states. If this switching is sufficiently fast then it is expected to “wash” out the frequency dependence that is observed between cells that start out ON versus OFF. To explore the consequences of this stochastic switching for the frequency dependence observed in our simple model (Fig 1), we performed simulations of a stochastic model that allows for switching between these states *with switching rates that are dependent upon the sugar concentration*, as follows:”

We have now heard back from the referee (previous reviewer #3) who thinks that the study has been largely improved after the inclusion of new data and supports its publication in *Molecular Systems Biology*. However, s/he still lists some concerns, which we would ask you to address in a revision.

 REFEREE COMMENTS

Reviewer #1:

I would like to thank the Authors for their efforts to address my comments. The manuscript has improved considerably. With the revisions and the new data on mutual invasibility in a constant environment, the manuscript should be acceptable for publication once the following few minor comments are addressed.

(1) The discussion about constant and changing environment shows the importance of clarifying the environmental conditions where observations are made. Therefore, I would like to suggest that statements of mutual invasibility, fitness optima or other features in the text should always specify the environments to which these statements refer. For example, "The simple foraging game predicts that altering the payoffs alters the stable equilibrium during resource depletion"; "Pure phenotypes are not mutually invisable in a bet-hedging scenario when the environment is constant or fluctuating" and so on.

(2) The discussion regarding my criticism of how the third criterion (fitness optimum) applies to the GAL system is really useful and informative. A shortened version of this discussion should be included in the main text, including the statement that "a non-optimal bet hedging strategy is still a bet-hedging strategy". Regarding the stability of non-optimal bet-hedging strategies, the Authors point out that non-optimal bet-hedging would be invisable by a mutant that implements a phenotypic mix bringing the population closer to the long-term growth optimum in a fluctuating environment. Two issues with this argument are: (i) natural fluctuating environments may be much more complex than a single stress or sugar ON/OFF at constant rates, meaning that bet-hedging populations may be trying to adapt to a constantly shifting optimum that they can never reach; (ii) even if the environment consists of only a single factor fluctuating ON/OFF, appropriate mutations may not be available for bringing the cells exactly to the fitness peak. Yes, the mutation space is practically infinite in general, but its breadth is less known if the goal is to bring the subpopulations and switching rates of a specific system (such as the GAL network) exactly to the optimum. This problem has not been experimentally explored, and any assumptions (such as the evolutionary accessibility of arbitrary population fractions and switching rates) may not hold true in reality. These additional points should also be part of the paragraph included in the main text.

(3) Some relevant citations to relevant works in the literature are missing, and should be included. Others are somewhat incorrect. Specifically, the beginning of the first paragraph in the introduction refers to bistability - but competence of cells in Reference 8 does not arise from bistability (the competence network in Ref. 8 is excitable, not bistable). Adaptive advantage of stochastic decision-making should have references to PMID 17176259 *Mol Microbiol.* 63(2):507-20 (2007); PMID: 19220745 *Mol Microbiol.* 71(6):1333-40 (2009); PMID: 17189188 *Mol Cell.* 24(6):853-65 (2006); PMID: PMC2851638 *Cell* 141(1): 69-80 (2010); and PMID 24045430 *Nat Commun.* 4:2467 (2013). Relevant references for evolution experiments are *Current Biology* 26, 1486-1493 (2016); PMID 26324468 *Mol Syst Biol.* 11(8):827 (2015); PMID 25255314 *PLoS Genet.* 10(10):e1004793 (2014); and *Nature Microbiology* (2016) doi: 10.1038/NMICROBIOL.2016.55.

Revision - authors' response

28 June 2016

Reviewer #1:

I would like to thank the Authors for their efforts to address my comments. The manuscript has

improved considerably. With the revisions and the new data on mutual invasibility in a constant environment, the manuscript should be acceptable for publication once the following few minor comments are addressed.

We thank the reviewer for the time and consideration he or she has put into reviewing this paper throughout multiple revisions. We have done our best to make the minor revisions requested, as detailed below:

(1) The discussion about constant and changing environment shows the importance of clarifying the environmental conditions where observations are made. Therefore, I would like to suggest that statements of mutual invasibility, fitness optima or other features in the text should always specify the environments to which these statements refer. For example, "The simple foraging game predicts that altering the payoffs alters the stable equilibrium during resource depletion"; "Pure phenotypes are not mutually invisable in a bet-hedging scenario when the environment is constant or fluctuating" and so on.

To address this request, we have attempted to make our claims more specific to the environmental conditions in which they apply (we have focused on statements regarding mixed strategy equilibria, since bet-hedging is specific to fluctuating environments). We have accordingly made changes to the following sections of the main text (changes in italics):

- The simple foraging game predicts that altering the payoffs alters the stable equilibrium *during resource depletion. (Line 201)*
- *Over the course of the resource depletion, the two pure strategies may thus be considered mutually invisable, with an equilibrium consisting of a stable mix of the two (Fig 1B) (Line 124)*
- Either way, *in that environment*, an isogenic population that adopts the stable mixed strategy via phenotypic heterogeneity renders itself uninvasible... (line 129)
- *Over the course of the resource depletion, however, at this growth-optimal fraction the individuals with phenotype B have lower fitness...*
- In the same context, we have also verified the theoretical prediction that the evolutionarily stable mixed strategy resulting from such frequency dependence is not necessarily optimal for population growth *over the course of resource depletion (line 324)*

(2) The discussion regarding my criticism of how the third criterion (fitness optimum) applies to the GAL system is really useful and informative. A shortened version of this discussion should be included in the main text, including the statement that "a non-optimal bet hedging strategy is still a bet-hedging strategy". Regarding the stability of non-optimal bet-hedging strategies, the Authors point out that non-optimal bet-hedging would be invisable by a mutant that implements a phenotypic mix bringing the population closer to the long-term growth optimum in a fluctuating environment. Two issues with this argument are: (i) natural fluctuating environments may be much more complex than a single stress or sugar ON/OFF at constant rates, meaning that bet-hedging populations may be trying to adapt to a constantly shifting optimum that they can never reach; (ii) even if the environment consists of only a single factor fluctuating ON/OFF, appropriate mutations may not be available for bringing the cells exactly to the fitness peak. Yes, the mutation space is practically infinite in general, but its breadth is less known if the goal is to bring the subpopulations and switching rates of a specific system (such as the GAL network) exactly to the optimum. This problem has not been experimentally explored, and any assumptions (such as the evolutionary accessibility of arbitrary population fractions and switching rates) may not hold true in reality. These additional points should also be part of the paragraph included in the main text.

We have added the following to our main text (changes italicized), starting line 394:

In our investigation of heterogeneity in the yeast GAL network, we have shown that frequency dependent foraging games are logical drivers of heterogeneity in metabolic networks. We have also shown that such interactions alone can drive evolution of phenotypic heterogeneity from a phenotypically homogenous population. *However, we do not rule out the possibility that such*

heterogeneity may also constitute a bet hedging or division of labor strategy under appropriate conditions of environmental uncertainty or kin/group selection. In particular, we have argued that because bet-hedging optima are theoretically growth-optimal for a population, the observation of our non-optimal stable mixed strategy is more consistent with the stable point of a negative frequency dependent game than with the stable point of a bet-hedging scenario (particularly given that clear frequency dependence exists between the phenotypes in a range of environments). Theoretically, a non-growth-optimal bet hedging strategy would be susceptible to invasion by a mutant that implements a phenotypic mix that provides a higher long-term population growth. The same is not true of a frequency-dependent game (for example, in the hawk-dove game an isogenic population following the “always-dove” strategy has the highest fitness but such a strategy nonetheless fails to invade the less-fit population composed of the stable mix of hawks and doves). However, we recognize that even a non-optimal bet-hedging strategy is still a bet-hedging strategy if over many environmental shifts the overall population fitness is higher with the phenotypic mix than with either phenotype alone. Furthermore, in complex environmental and metabolic scenarios there are many reasons for which the optimal bet-hedging strategy may be evolutionarily inaccessible to a population in an uncertain environment. We cannot therefore rule out bet-hedging as a possible co-contributor to the observed phenotypic heterogeneity in the yeast galactose network.

(3) Some relevant citations to relevant works in the literature are missing, and should be included. Others are somewhat incorrect. Specifically, the beginning of the first paragraph in the introduction refers to bistability - but competence of cells in Reference 8 does not arise from bistability (the competence network in Ref. 8 is excitable, not bistable).

Adaptive advantage of stochastic decision-making should have references to PMID 17176259 Mol Microbiol. 63(2):507-20 (2007); PMID: 19220745 Mol Microbiol. 71(6):1333-40 (2009); PMID: 17189188 Mol Cell. 24(6):853-65 (2006); PMID: PMC2851638 Cell 141(1): 69-80 (2010); and PMID 24045430 Nat Commun. 4:2467 (2013).

Relevant references for evolution experiments are Current Biology 26, 1486-1493 (2016); PMID 26324468 Mol Syst Biol. 11(8):827 (2015); PMID 25255314 PLoS Genet. 10(10):e1004793 (2014); and Nature Microbiology (2016) doi: 10.1038/NMICROBIOL.2016.55.

We thank the reviewer for the careful reading of our citations and suggestions for further reference. We have removed the specification of bistability in the introduction, which now reads:

In some cases, stochastic gene expression results in the coexistence of distinct phenotypes among genetically identical cells [5,6].

We have also added the references suggested by the reviewer.

Corresponding Author Name: Jeff Gore

Manuscript Number: MSB-16-7033